# LEARN-BY-INTERACT: A DATA-CENTRIC FRAMEWORK FOR SELF-ADAPTIVE AGENTS IN REALISTIC ENVIRONMENTS

**Hongjin Su** [12] , **Ruoxi Sun** [1] , **Jinsung Yoon** [1] , **Pengcheng Yin** [1] , **Tao Yu** [2] , **Sercan Ö. Arık** [1]

[1] Google , [2] The University of Hong Kong

## ABSTRACT

Autonomous agents powered by large language models (LLMs) have the potential to enhance human capabilities, assisting with digital tasks from sending emails to performing data analysis. The abilities of existing LLMs at such tasks are often hindered by the lack of high-quality agent data from the corresponding environments they interact with. We propose LEARN-BY-INTERACT, a data-centric framework to adapt LLM agents to any given environments without human annotations. LEARN-BY-INTERACT synthesizes trajectories of agent-environment interactions based on documentations, and constructs instructions by summarizing or abstracting the interaction histories, a process called *backward construction*. We assess the quality of our synthetic data by using them in both training-based scenarios and training-free in-context learning (ICL), where we craft innovative retrieval approaches optimized for agents. Extensive experiments on SWE-bench, WebArena, OSWorld and Spider2-V spanning across realistic coding, web, and desktop environments show the effectiveness of LEARN-BY-INTERACT in various downstream agentic tasks — baseline results are improved by up to 11.1% for ICL with Claude-3.5 and 23.1% for training with Codestral-22B. We further demonstrate the critical role of backward construction, which provides up to 10.6% improvement for training. Our ablation studies demonstrate the efficiency provided by our synthesized data in ICL and the superiority of our retrieval pipeline over alternative approaches like conventional retrieval-augmented generation (RAG). We expect that LEARN-BY-INTERACT will serve as a foundation for agent data synthesis as LLMs are increasingly deployed at real-world environments.

## 1 INTRODUCTION

Pre-trained large language models (LLMs) offer great potential for assisting humans with various tasks in digital settings, such as editing images, performing data analysis, resolving software engineering issues, and navigating commercial platforms (Xie et al., 2023; 2024; Yao et al., 2022a; Jimenez et al., 2023). By streamlining these, LLM agents can greatly enhance human efficiency and productivity, allowing individuals to shift their focus toward higher-level, creative, and strategic endeavors. To explore this potential, many benchmarks (Jimenez et al., 2023; Zhou et al., 2023b; Xie et al., 2024; Cao et al., 2024; Koh et al., 2024) and agentic frameworks (Yang et al., 2024; Zhan & Zhang, 2023; Yang et al., 2023; Gur et al., 2023; Chen et al., 2024a) have been established based on realistic digital environments, spanning web applications, code development, desktop computing, etc. However, current LLMs often fall short of expected performance in these tasks, consistently displaying a significant gap compared to human capabilities. As a result, they remain less practical and reliable for real-world applications.

Efficient adaptation to new environments can be the key part of the performance improvements. Prior works have explored various prompt-based approaches (Yao et al., 2022b; Yang et al., 2024; Gur et al., 2023; Zhan & Zhang, 2023), that are constrained by the capabilities of underlying foundation models. Other studies on training LLMs with human-labeled examples (Chen et al., 2023; 2024b; Li et al., 2020) on the other hand, come with the fundamental limitation of high annotation costs when new environments are considered. In particular, annotating agentic data can be quite

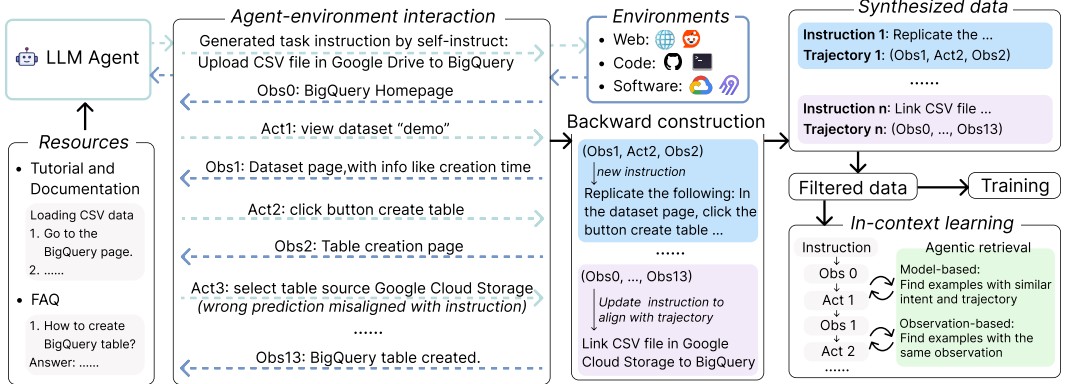

Figure 1: Overview of the data synthesis and adaptation processes. Given an environment and standard resources, we first leverage self-instruct to create a diverse set of instructions. LLMs are then employed to complete these tasks, resulting in long trajectories of agent-environment interactions. We construct task instructions using LLMs for each sub-trajectory, a process called *backward construction*. The synthesized data are then filtered and used for both training and in-context learning, where we design agentic retrieval to retrieve demonstration examples based on information at each step, using both model-based and observation-based approaches. See Appendix F for the complete data synthesis example and Algorithm 2 for more details on agentic retrieval.

difficult and expensive due to long-trajectory interactions with environments and specific domain expertise required. Few works have explored fully-autonomous data construction pipelines towards self-adaptive agents that can efficiently learn new environments (Gulcehre et al., 2023; Aksitov et al., 2023).

In this paper, we introduce LEARN-BY-INTERACT, a data-centric framework for LLMs to self-adapt to new environments, utilizing agent data synthesis via interactions. Intuitively, the effects of actions executed in environments (e.g., the next webpage after clicking a button) serve as informative demonstrations that help LLMs in future navigation. Inspired by this, we design LEARN-BY-INTERACT that first uses self-instruct (Wang et al., 2022b) to develop a variety of task instructions, referring to standard resources such as documentations and tutorials for a given environment. This covers most important scenarios that human users are interested in and avoids intensive prompt engineering to control the distribution and diversity of the generated data. We then collect diverse trajectories from interactions between LLMs and environments, as illustrated in Fig. 1. However, given the low performance of LLMs in existing agentic benchmarks (Xie et al., 2024; Cao et al., 2024), it is likely that a large percentage of synthesized trajectories do not match with the instructions. To tackle this challenge, we construct new instructions by summarizing or abstracting each sub-trajectory, leveraging the strong summarization capabilities of LLMs (Pu et al., 2023; Liu et al., 2023). We call this process *backward construction*. After obtaining synthesized instruction-trajectory pairs and filtering low-quality ones, we apply it to both training and ICL, where we craft innovative retrieval pipelines optimized for agents. Concretely, it consists of two parts: (1). model-based approach that leverages LLMs to first write queries based on instructions, interaction histories and current observations, and uses retrieval models to retrieve demonstration examples from synthesized data; (2). observation-based approach that finds examples with the current observation appearing in trajectories (which indicates that the current state has been encountered in the data synthesis process).

Our comprehensive evaluations across four challenging benchmarks: SWE-bench (Jimenez et al., 2023), WebArena (Zhou et al., 2023b), OSWorld (Xie et al., 2024), and Spider2-V (Cao et al., 2024), highlight the efficacy of the data generated by LEARN-BY-INTERACT. With ICL, both Gemini-1.5-pro (Reid et al., 2024) and Claude-3.5-sonnet (Anthropic, 2024) show consistent and remarkable improvements – for OSWorld (Xie et al., 2024), our generated data nearly doubles Claude-3.5-sonnet's baseline performance, increasing it from 11.4% to 22.5%. Furthermore, substantial improvements are observed by training models of varying sizes and architectures with our synthesized data. As an example, Codestral-22B's (Team, 2024b) performance in WebArena significantly increases from

4.7% to 27.8% after training. These results underscore the high quality of our generated agent data and its broad applicability across diverse agent environments.

Our extensive ablation studies reveal that backward construction not only increases the quantity of the synthesized data, but also improves its overall quality (§3.5). With data synthesized by LEARN-BY-INTERACT, we observe significant improvements in both performance and efficiency during LLM inference (§4.1). Our empirical results demonstrate the superiority of the agentic retrieval in ICL (§4.2). We anticipate that this research will spark innovative developments in enhancing agent performance using LLMs and contribute to its wider-spread adoption in real-world application scenarios.

## 2 LEARN-BY-INTERACT

We introduce the LEARN-BY-INTERACT pipeline to synthesize agent data in an autonomous way by leveraging interactions between LLMs and environments. We first formalize the agent canonical task (§2.1), and introduce the detailed synthesis (§2.2) and filtering (§2.3) procedures. We then describe the application of the synthesized data in adapting LLMs in both training-free and training-based settings (§2.4).

### 2.1 TASK FORMULATION

Given an environment $E$ and a task instruction $I$, the objective of an agent $A$ is to achieve the target $G$ through multi-step interactions with $E$. At each step $i$, $A$ predicts the next action $a_i$ based on the instruction $I$ and the previous history $H = (o_0, a_1, o_1, a_2, ..., o_{i-1})$, which is then executed in the environment $E$ to get a new observation $o_i$. The interactions terminated until $A$ predicts the action $stop$ or the maximum number of steps $m$ is reached.

### 2.2 AGENTIC DATA SYNTHESIS

The essential idea of LEARN-BY-INTERACT is manifested in synthesizing environment-specific agent data with zero human effort. In Algorithm 1, we show the overall process with pseudo-code. Given an environment for a downstream application (such as visual studio code), we first leverage commonly-accessible resources like documentation to generate diverse task instructions using self-instruct (Wang et al., 2022b) (line 5). These resources are usually created by human experts to address common concerns and provide usage suggestions, e.g., how to navigate a website or operate a software. Intuitively, such references often cover representative usecases of an application. Therefore the task instructions generated conditioned on them could cover most popular scenarios in the domain and avoids potentially unreasonable cases that may be of less value.

For each generated task, LLMs then aim to solve it, which results in a long trajectory $T = (o_0, a_1, o_1, ..., a_n, o_n)$ (line 9-14 in Algorithm 1). To address the potential misalignment between the instruction $I$ and the generated trajectories $T$, we introduce a novel mechanism, backward construction, to construct instructions based on trajectories (line 15-22 in Algorithm 1). Specifically, for each sub-trajectory

---

**Algorithm 1** Agent data synthesis

1: **Input:** $LLM$: Large Language Model; $E$: environment; $Doc$: standard resources like documentation; $N$: the number of instructions to generate per document; $F$: data filter.
2: **Initialization:** $D = []$: synthesized data.
3: **for** $d$ in $Doc$ **do**
4:     // self-instruct to generate $N$ task instructions
5:     $Instructions = LLM(d, N)$
6:     **for** $I$ in $Instructions$ **do**
7:         $E$.reset()
8:         $T = []$    // initialize interaction trajectory
9:         **while** not $E$.finished() **do**
10:             $o = E$.get_observation()
11:             $a = LLM(I, T, o)$
12:             $T$ += $[o, a]$
13:         **end while**
14:         $T.append(E$.get_observation())
15:         // backward construction
16:         **for** $i$ in range$(0, len(T) - 1, 2)$ **do**
17:             **for** $j$ in range$(i + 2, len(T), 2)$ **do**
18:                 $T' = T[i : j]$
19:                 $I' = LLM(T')$
20:                 $D.append([I', T'])$
21:             **end for**
22:         **end for**
23:     **end for**
24: **end for**
25: $D = F(D)$ // Filter low-quality data
26: **Return:** $D$

---

$T' = (o_i, a_{i+1}, o_{i+1}, ..., a_j, o_j), 0 \le i < j \le n$, we obtain two types of new instructions: (1). summarizations of trajectory steps; (2). abstractions of the trajectory purpose. In Fig. 1, the sub-trajectory $(Obs1, Act2, Obs2)$ is summarized into a new task instruction that requires to replicate the $Act2$. The abstraction of the full trajectory updates the original task objective, which is no longer aligned with the generated trajectory due to the wrong prediction in the action 3. Overall, the LEARN-BY-INTERACT pipeline offers two notable advantages:

- It corrects the potential misalignment between instructions and predicted trajectories by updating task objectives, which enhances the data quality as verified by the experimental results in §3.5.

- It maximizes the utility of each generated trajectory by crafting new instructions for each sub-trajectory. This results in a quadratic increase in the number of synthesized examples with respect to the steps in the sequence per generated trajectory. For a given target dataset size, backward construction substantially decreases the necessary interactions, which is particularly valuable in scenarios where such interactions are challenging and costly to obtain such as Robotics (Keipour, 2022).

### 2.3 FILTERING

To further enhance the data quality, we design the following criteria to filter inferior synthesized data: (1). Remove duplicate states: We remove duplicate $(a_i, o_i)$ from $T'$ if $(a_i, o_i)=(a_{i-1}, o_{i-1})$, which is potentially introduced by the invalid action or the environment error (inactivity). (2). LLM committee check: We feed the generated instruction-trajectory pair $(I', T')$ into a committee of LLMs, and only classify it of high-quality if all LLMs consider the trajectory coherent, natural, reasonable and aligned with the instruction. The listed criteria are all fully-autonomous and canonically-applicable for filtering data synthesized in general agent scenarios. See Table 35 for our prompts used in LLM committee check.

### 2.4 ADAPTATION

After obtaining the synthesized data $D$, we apply it to both ICL and training. Given the unique characteristics of multi-round interactions with environments in agent settings, we design agentic retrieval (pseudo-code in Algorithm 2) to maximize the effectiveness of the synthesized data. Specifically, we propose two retrieval pipelines: observation-based (line 5-14) and model-based retrieval (line 15-17). In observation-based retrieval, we compare the current observation $o$ to the trajectory of each example $e$ in the synthesized data, where $e = [I', [o_0, a_1, o_1, ..., a_n, o_n]]$. If $o$ matches one of the observations in $e$, i.e., $o = o_i$, then we consider $e$ as a helpful example to the current task. For the model-based retrieval, we leverage LLMs to first write queries based on the instruction, the interaction history and the current observation (line 16), and then employ retrieval models to retrieve non-duplicate examples (line 17). LLMs are then augmented with the retrieved examples to predict the next action (line 18). Refer to Table 36 to 39 for prompts to write queries and predict actions.

---

**Algorithm 2** ICL with agentic retrieval

1: **Input:** $LLM$: Large Language Model; $E$: environment; $D$: synthesized data; $RM$: retriever; $I$: task instruction; $m1$: maximum number of examples from observation-based retrieval; $m2$: maximum number of examples from model-based retrieval.
2: **Initialization**: $H = []$: interaction history; $R$: retrieved examples.
3: **while** not $E$.finished() **do**
4:     $o = E$.get_observation()
5:     // observation-based retrieval
6:     **for** $i, t$ in $D$ **do**
7:         // iterate through the trajectory
8:         **for** $o_1$ in $t$ **do**
9:             **if** $o_1=o$ **then**
10:                 $R$.append($[i, t]$)
11:             **end if**
12:         **end for**
13:     **end for**
14:     $R = R[:m1]$
15:     // model-based retrieval
16:     $q = LLM(I, H, o)$
17:     $R \mathrel{+}= RM(q, D, m2, R)$
18:     $a = LLM(I, H, o, R)$
19:     $H \mathrel{+}= [o, a]$
20: **end while**

---

Apart from using the synthesized data as demonstration examples in ICL, we further utilize them to fine-tune models. For a given generated example, we convert it to the format of action prediction (Table 36), and prepare input-output pairs for supervised fine-tuning. More details on the experimental settings can be found in §3.3.

Table 1: Statistics for the number of crawled documents, generated raw trajectories, examples (instruction-trajectory pairs) and examples after filtering.

|  | SWE-bench | WebArena | OSWorld | Spider2-V |
|---|---|---|---|---|
| Documents | 6,464 | 3,578 | 7,362 | 11,231 |
| Raw trajectories | 19,392 | 10,734 | 22,086 | 33,693 |
| Examples | 180,752 | 185,635 | 437,635 | 652,786 |
| Filtered examples | 101,523 | 109,276 | 103,526 | 125,683 |

## 3 EXPERIMENTS

### 3.1 BASELINES

We compare ICL with agentic retrieval to the following prompt-based approaches.

- Baseline: The vanilla prediction pipeline in each benchmark that includes the task instruction, interaction history and the state observation in the prompt. See more implementation details in Appendix A.
- RAG: The conventional RAG pipeline that first retrieves from the resources like documentation based on the instruction, and augments LLMs with the retrieved content.
- Data distill: We follow the same pipeline to synthesize data in Algorithm 1 except backward construction (replace line 15-22 with $D.append(I, T)$), and follow Algorithm 2 during the evaluation.
- Reflexion (Shinn et al., 2024): A general framework to reinforce language agents through linguistic feedback from both executors and LLMs.
- Language Agent Tree Search (LATS) (Zhou et al., 2023a): It integrates the combinatorial tree search into expanding ReAct (Yao et al., 2022b) and combine agent online reasoning, acting and planning throughout the trajectory.

For the training-based evaluation, we primarily compare to the data distillation, which also constructs data from scratch and requires no human effort to annotate seed or preference data. Additionally, we include the model performance before training as another baseline.

### 3.2 DATASETS

We consider 4 agent datasets that involve multi-round interactions with realistic environments. They span diverse domains of code, web, computer desktop and professional software. Appendix C illustrates details of each dataset with examples.

- SWE-bench (Jimenez et al., 2023) is an evaluation benchmark on realistic software engineering problems from realistic Github issues. We use the Lite version by default throughout the experiments.
- Webarena (Zhou et al., 2023b) evaluates agent capabilities to perform tasks in the web environments such as e-commerce, social forum discussion, and beyond.
- OSWorld (Xie et al., 2024) is an integrated environment for assessing open-ended computer tasks, which involve diverse applications like terminal, chrome, etc.
- Spider2-V (Cao et al., 2024) is a multimodal agent benchmark focusing on professional data science and engineering workflows, which includes BigQuery, Airbyte and more.

### 3.3 SETTINGS

We synthesize one separate set of environment-specific data for each evaluated benchmark. Throughout the data synthesis process, we employ the Claude-3.5-sonnet (Anthropic, 2024) as the generator model and both Gemini-1.5-pro (Reid et al., 2024) and Claude-3.5-sonnet as the LLM committee for filtering low-quality data. For each document, we sample three task instructions from

Table 2: Comparison of LEARN-BY-INTERACT to other existing training-free approaches. SWE refers to SWE-bench, Web refers to WebArena and OS refers to OSWorld. The best results are highlighted in bold. We include more leaderboard results of SWE-bench and WebArena in Table 6.

| Benchmark → | SWE | Web | OS | Spider2-V | SWE | Web | OS | Spider2-V |
|---|---|---|---|---|---|---|---|---|
| Approach ↓ | | Gemini-1.5-pro | | | | Claude-3.5-sonnet | | |
| | | *Existing approaches* | | | | | | |
| Baseline | 13.3 | 17.9 | 4.9 | 8.3 | 26.7 | 31.5 | 11.4 | 7.5 |
| RAG | 13.7 | 19.5 | 5.1 | 9.1 | 27.0 | 31.8 | 11.7 | 7.7 |
| Data distill | 14.0 | 19.8 | 5.7 | 9.1 | 28.0 | 32.1 | 11.9 | 8.5 |
| Reflexion | 14.3 | 20.2 | 5.7 | 9.3 | 28.3 | 32.4 | 12.2 | 8.9 |
| LATS | 15.3 | 21.0 | 6.5 | 11.3 | 29.0 | 34.2 | 13.6 | 10.3 |
| | | *Ours* | | | | | | |
| Learn-by-interact | **18.7** | **25.6** | **10.3** | **16.4** | **34.7** | **39.2** | **22.5** | **16.3** |
| Δ over baseline | +5.4 | +7.7 | +5.4 | +8.1 | +8.0 | +7.7 | +11.1 | +8.8 |

LLMs. The statistics for generated raw trajectories, examples before and after filtering are shown in Table 1. In Appendix E, we list document sources used for each benchmark. During ICL, we retrieve examples until the maximum length of LLMs and set an upper bound of 5 for both model-based and observation-based retrieval ($m1 = 5$, $m2 = 5$ in Algorithm 2). We leverage Gemini-1.5-pro (Reid et al., 2024) and Claude-3.5-sonnet (Anthropic, 2024)[1], Codegemma-7B (Team, 2024a) and Codestral-22B (Team, 2024b) in the ICL evaluation, and tune Codegemma-7B and Codestral-22B with LoRA (Hu et al., 2021) to evaluate the data quality as training sources. By default, we do not include retrieval content in evaluating the trained model to avoid the confusion in understanding the effectiveness of our synthesized data in training. We include more detailed hyper-parameter settings (both existing approaches and LEARN-BY-INTERACT) and machine information in Appendix D.

## 3.4 EVALUATION

We follow the default evaluation metrics designed by the original benchmarks. In SWE-bench (Jimenez et al., 2023), we apply the generated patch program to the repository codebase, and measure the agent performance by execution accuracy (pass@1). In WebArena (Zhou et al., 2023b), we employ both LLM-based fuzzy match and string match that checks keywords in predictions. Slightly different from the original work that uses gpt-4-0613 as the LLM judge, we use Claude-3.5-sonnet as a similar replacement. In OSWorld (Xie et al., 2024), we leverage the sample-specific evaluation scripts to assess the functional correctness of the task completion, which processes environment states and checks if agents finish the task as expected. In Spider2-V (Cao et al., 2024), we utilize file-based comparison, information-based validation, execution-based verification to determine whether a task is successfully completed. All performance numbers throughout the paper are shown in the percentage of resolved instances with % omitted for brevity.

## 3.5 RESULTS

### 3.5.1 TRAINING-FREE EVALUATION

We first consider LEARN-BY-INTERACT in the training-free setting, where the proposed methods can be applied to the commercial LLMs even with prediction-only API access.

Results on Table 2 show marginal improvement of RAG compared to the baseline, which suggests limited effectiveness by simply concatenating standard reousrces to LLM prompts. By retrieving examples from distilled data, we observe better performance compared to RAG, but still no more than 2% improvement over the baseline, which indicates that the distilled data tend to be noisy in the setting with multi-round agent-environment interactions. This highlights the critical role of

---

[1]In the subsequent descriptions, Gemini refers to Gemini-1.5-pro, and Claude refers to Claude-3.5-sonnet.

Table 3: Downstream task performance of models trained from data generated by Learning-by-interact and data distillation. We include the models results before training, where the synthesized data is used as demonstration examples, and after training, where the synthesized data is used to train models.

| Benchmark → | Web | OS | Web | OS | Web | OS | Web | OS |
|---|---|---|---|---|---|---|---|---|
| Model → | Codegemma-7B | | Codestral-22B | | Codegemma-7B | | Codestral-22B | |
| Approach ↓ | *Before tuning* | | | | *After tuning* | | | |
| | *Existing approaches* | | | | | | | |
| Baseline | 3.3 | 0.0 | 4.7 | 2.2 | - | - | - | - |
| Data distill | 4.2 | 0.0 | 5.8 | 2.7 | 6.2 | 1.4 | 10.2 | 5.4 |
| | *Ours* | | | | | | | |
| Learn-by-interact | 7.6 | 3.5 | 9.9 | 5.4 | 17.9 | 6.5 | 27.8 | 11.7 |
| Δ over baseline | +4.3 | +3.5 | +5.2 | +3.2 | +14.5 | +6.5 | +23.1 | +9.5 |

backward construction, which corrects the misalignment between instructions and trajectories by curating new task objectives.

Both Reflexion and LATS consistently improve over the baseline across 4 benchmarks, which demonstrate their general applicability to agent tasks. Using the data synthesized from the LEARN-BY-INTERACT, we can see a significant performance gain compared to all other frameworks in both Gemini and Claude. For example, in OSWorld, augmenting Claude with synthesized environment-specific data almost doubles the result compared to the baseline. This signifies the high quality of the generated data and the effectiveness of the LEARN-BY-INTERACT framework.

### 3.5.2 TRAINING-BASED EVALUATION

We consider the data synthesized by LEARN-BY-INTERACT in the scenario of LLM tuning, which is applicable to the LLMs with access to weight updates.

The results presented in Table 3 reveal that LEARN-BY-INTERACT substantially surpasses both the baseline and data distillation, suggesting its capacity to generate high-quality training data that enables language models to learn and adapt efficiently. We discover that utilizing our synthesized data for model training yields better results compared to using it as in-context learning (ICL) examples. A notable instance is in WebArena, where Codestral-22B's performance jumps from 4.7% to 27.8% when trained on our synthesized data, while only showing a 5.2% improvement in the ICL scenario. Remarkably, the Codestral-22B model trained with our synthesized data even outperforms Gemini when the latter uses our data as demonstration examples.

## 4 ANALYSIS

### 4.1 INFERENCE EFFICIENCY

We compare the efficiency of different pipelines at inference. We analyze the trade-off between downstream task performance and the required computational costs. We focus on measuring the number of LLM calls and consumed tokens per example, which are averaged across four evaluated datasets (§3.2) using Claude-3.5-sonnet. As illustrated in Fig. 2, while Reflexion and LATS demonstrate enhanced performance, this comes at the cost of significantly increased computational resources during inference. Specifically, LATS yields a 2.5% improvement on average, but requires nearly four times used tokens per instance relative to the baseline. In contrast, LEARN-BY-INTERACT exhibits superior performance while utilizing fewer LLM calls and slightly more tokens compared to the baseline. Thanks to the rich environment information stored in the examples of synthesized data, LLMs can potentially make better decisions and thus finish the task in fewer steps. This removes the performance-efficiency trade-off during inference at the cost of data synthesis in

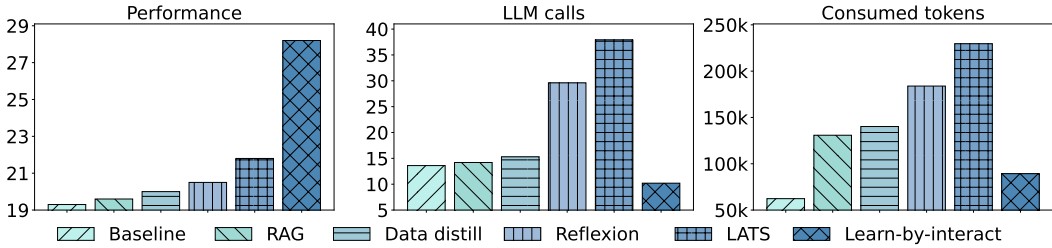

Figure 2: Evaluation performance, the number of LLM calls and consumed tokens (per example) of various training-free pipelines during inference, which are all averaged across four benchmarks: SWE-bench, Webarena, OSWorld and Spider2-V.

Table 4: Model performance based on different retrieval paradigms. Observation-based and Model-based retrieval prove to be particularly effective in agent tasks, whose combination (ours) gives the best results.

| Benchmark → | SWE | Web | OS | Spider2-V | SWE | Web | OS | Spider2-V |
|---|---|---|---|---|---|---|---|---|
| Retrieval ↓ | | Gemini-1.5-pro | | | | Claude-3.5-sonnet | | |
| No retrieval | 13.3 | 17.9 | 4.9 | 8.3 | 26.7 | 31.5 | 11.4 | 7.5 |
| Instruction-based | 14.7 | 21.6 | 7.0 | 10.2 | 27.7 | 33.6 | 15.7 | 9.1 |
| Observation-based | 16.3 | 23.5 | 8.7 | 14.6 | 32.3 | 36.3 | 18.7 | 13.2 |
| Model-based | 17.0 | 24.3 | 9.5 | 15.4 | 33.7 | 37.2 | 20.3 | 14.5 |
| Ours | 18.7 | 25.6 | 10.3 | 16.4 | 34.7 | 39.2 | 22.5 | 16.3 |

advance and suggests that LEARN-BY-INTERACT is particularly well-suited for real-world deployment that demonds both low latency and high performance.

## 4.2 THE IMPACT OF RETRIEVAL

As mentioned in §2.4, we employ both model-based and observation-based retrieval in our evaluation with ICL. We analyze their effectiveness by incorporating only one of them (skip line 5-14 in Algorithm 2 for model-based retrieval only and skip line 15-17 for observation-based retrieval only). In addition, we compare to two baselines: (1). no retrieval: LLMs predict each action in the zero-shot setting; (2). instruction-based: only use instructions to retrieve synthesized data and apply the same demonstration examples in every action prediction throughout the trajectory.

The results presented in Table 4 illustrate how various retrieval methods impact LLMs when using the synthetic data as the retrieval source. Despite having access to the same example pool (except the baseline without using retrieval), there are notable differences in performance across different retrieval strategies, highlighting the crucial role of agentic retrieval in effectively utilizing synthesized data. Traditional Retrieval-Augmented Generation (RAG) methods, which only employs instructions for retrieval, show the least improvement across four benchmarks and two LLMs. In contrast, the observation-based approach proves particularly effective for agent-based tasks, significantly outperforming the instruction-based retrieval, for instance, achieving a 4.4% absolute improvement in Spider-2V when using Gemini. By leveraging task instructions, interaction history and the current observation, model-based retrieval demonstrates even better results compared to using the observation-based version. Ultimately, the most impressive scores are achieved by combining both model-based and observation-based retrieval, which results in our agentic retrieval pipeline. These findings underscore the importance of carefully designing retrieval pipelines to maximize the potential of synthetic data and LLMs in agent scenarios.

## 4.3 DATA GRANULARITY

Table 5: Effectiveness of synthetic data with various granularity. In general, short-trajectory data is more advantageous to both training and ICL, while mixing all of short, medium and long-trajectory data provides the best performance.

| Benchmark → | SWE | Web | OS | Spider2-V | Web | OS |
|---|---|---|---|---|---|---|
| Granularity ↓ | | Claude-3.5-sonnet | | | Codestral-22B | |
| Baseline | 26.7 | 31.5 | 11.4 | 7.7 | 4.6 | 2.2 |
| Short | 28.7 | 33.3 | 14.9 | 10.3 | 13.5 | 4.9 |
| Medium | 28.0 | 32.5 | 13.8 | 9.5 | 12.6 | 4.0 |
| Long | 27.3 | 31.9 | 13.0 | 8.9 | 10.6 | 3.4 |
| Short+Medium | 30.0 | 34.4 | 15.7 | 10.7 | 14.6 | 5.7 |
| Short+Long | 29.3 | 33.9 | 15.2 | 10.5 | 14.4 | 5.3 |
| Medium+Long | 28.7 | 32.9 | 14.4 | 10.1 | 13.2 | 4.5 |
| Short+Medium+Long | 31.0 | 34.9 | 16.3 | 11.3 | 15.4 | 6.3 |

As mentioned in §2.2, we synthesize data by taking contiguous sub-trajectories from the full generation paths of LLMs, i.e. $T' = T[i : j]$, which results in trajectories of diverse lengths in the synthesized data. We divide the synthetic data into three groups: (1). trajectory steps $< 5$ (short); (2). $5 \leq$ trajectory steps $< 10$ (medium); (3). trajectory steps $\geq 10$ (long), and leverage each group and their combinations in both the training-free and the training-based process. To ensure a fair comparison, we constraint the data size in each group and combined group to 200M tokens[2], utilizing Su et al. (2022) for sub-sampling. Table 5 presents the results. In both training-free and training-based evaluation, LLMs derive greater advantages from short-trajectory data, as demonstrated by its consistently superior performance compared to medium and long-trajectory data with Claude-3.5-sonnet and Codestral-22B. This can be at-

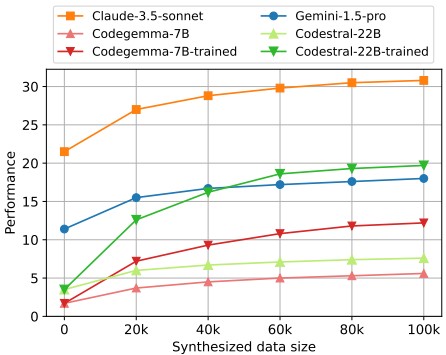

Figure 3: Scaling laws of the synthesized data. Compared to in-context learning, tuned models achieves more significant improvements as the data scales up. The performance is averaged across WebArena and OSWorld.

tributed to the versatility of short-trajectory data, which usually serves as a sub-step or a partial workflow in downstream tasks. The combination of any two data groups proves more effective than relying on a single group, showcasing the complementary nature of diverse data sets. For instance, in Webarena with Codestral-22B, incorporating examples with both short and medium-length trajectories shows additional improvement over using either one exclusively (14.6 vs 13.5 and 14.6 vs 12.6). This underscores the value of considering the trajectory length as a unique dimension of agent data synthesis.

## 4.4 SCALING LAWS

We examine how the model performance improves as the synthetic data size scales up. Figure 3 presents two sets of results, with training-free (where Claude, Gemini, Codegemma and Codestral use retrieval augmentation without training) and with training-based (where fine-tuned Codegemma and Codestral models are evaluated without retrieval). All results are averaged across Webarena and OSworld due to the limit of computational resources. The findings indicate that both learning paradigms benefit from larger data, suggesting the synthetic data is diverse and high-quality. In the training-free evaluation, more substantial improvements are observed for larger models (Claude and Gemini) compared to smaller ones (Codegemma and Codestral), possibly due to the enhanced

---

[2]We use the number of tokens to measure the data size due to the fact that long-trajectory example may contain more information compared to the short version.

in-context learning abilities of larger models. Our analysis also reveals that for a given amount of synthetic data, fine-tuning smaller models is more effective than using the data as demonstration examples during evaluation.

## 5 RELATED WORK

Various agents based on LLMs have been developed (Wang et al., 2024; Zhang et al., 2024; Shinn et al., 2024; Huang et al., 2022; Wang et al., 2023a;b). React (Yao et al., 2022b) proposes to synergize reasoning and acting in LLMs. By integrating Monte Carlo Tree Search (Kocsis & Szepesvári, 2006; Coulom, 2006), Zhou et al. (2023a) leverages LLM-powered value functions and self-reflection (Madaan et al., 2024) to encourage proficient exploration and decision-making. However, it comes with increased computational costs and relies on the premise that the environment allows for state reversals. In contrast, LEARN-BY-INTERACT removes such assumptions and improves both agent efficiency and performance by synthesizing high-quality data in advance.

Another line of research to improve agent models relies on training on human-labeled examples (Zeng et al., 2023; Yin et al., 2023; Deng et al., 2024; Chen et al., 2024b; Wang et al., 2022a) or data distilled from LLMs like GPT-4 (Chen et al., 2023; Zhao et al., 2024). AgentGen (Hu et al., 2024) explores automatic synthesis of both environments and tasks and then leverages FastDownward[3] to generate trajectory data. AgentTuning (Zeng et al., 2023) utilizes both existing datasets and self-instruct (Wang et al., 2022b) to derive instructions and then samples trajectories from GPT-4 (Achiam et al., 2023). In contrast, LEARN-BY-INTERACT focuses on realistic environments and generate tasks and trajectories using backward construction. Some other researchers are also exploring ways to use data more efficiently with reinforcement learning (Ball et al., 2023; Schwarzer et al., 2020; Nachum et al., 2018; Thomas & Brunskill, 2016; Schwarzer et al., 2021). Gulcehre et al. (2023) suggests using data created by an LLM's policy can enhance itself via offline reinforcement learning algorithms. Aksitov et al. (2023) takes this further by combining with ReAct (Yao et al., 2022b) to train agent models iteratively on experience trajectories. These typically require a reward model as the scoring function or LLM/execution-generated feedback to enhance data quality. Our work, however, takes a different approach by employing the backward construction to improve the data quality by aligning instructions and trajectories.

## 6 CONCLUSION

We introduce LEARN-BY-INTERACT, a data-centric framework to adapt LLM agents to any given environments without human annotations. Based on commonly-accessible resources like documentaion, LLMs propose downstream tasks and complete them with multi-round interactions with environments. We address the misalignment between instructions and trajectories by updating objectives with new instructions derived from trajectories. Additionally, we design innovative retrieval pipelines that leverage agent instructions, interaction histories, and current observations to retrieve synthesized examples. Through extensive experiments, we demonstrate that the synthetic data from LEARN-BY-INTERACT significantly enhances model performance in ICL and training. Compared with other leading approaches in agent tasks, LEARN-BY-INTERACT shows much better performance with lower latency and computational costs, which make it particularly suitable for large-scale deployment. Further analysis has also shown the superiority of LEARN-BY-INTERACT over the classical RAG. In future work, we plan to explore multi-modal settings and train general agent models widely applicable in realistic environments. We anticipate that LEARN-BY-INTERACT will inspire future research to push the state-of-the-art in this direction.

## 7 LIMITATIONS

Although LEARN-BY-INTERACT effectively synthesizes high-quality agentic data with trajectories, it requires a lot of LLM calls in generation and filtering. We hope that future works will explore more efficient approaches to complete annotations without sacrificing quality. Additionally, LEARN-BY-INTERACT leverages the environment-related resources to generate instructions. In some scenarios, however, these resources may be incomplete or not available.

---

[3]https://www.fast-downward.org/

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

Table 6: Top-10 results of SWE-bench from the leaderboard at `https://www.swebench.com/`. All the numbers are fetched on 2024-10-01.

| Approach ↓ | site | result |
|---|---|---|
| CodeStory Aide + Mixed Models | https://www.swebench.com/ | 43.0 |
| Honeycomb | https://honeycomb.sh/ | 38.3 |
| AbanteAI MentatBot | https://mentat.ai/blog/mentatbot-sota-coding-agent | 38.0 |
| Gru | https://gru.ai/ | 35.7 |
| Isoform | https://www.isoform.ai/ | 35.0 |
| SuperCoder2.0 | https://superagi.com/supercoder/ | 34.0 |
| MarsCode | https://www.marscode.com/ | 34.0 |
| Lingma | https://arxiv.org/abs/2406.01422 | 33.0 |
| Factory Code Droid | https://www.factory.ai/ | 31.3 |
| AutoCodeRover | https://autocoderover.dev/ | 30.7 |
| LEARN-BY-INTERACT (ours) | This paper | 34.7 |

Shuyan Zhou, Frank F Xu, Hao Zhu, Xuhui Zhou, Robert Lo, Abishek Sridhar, Xianyi Cheng, Tianyue Ou, Yonatan Bisk, Daniel Fried, et al. Webarena: A realistic web environment for building autonomous agents. *arXiv preprint arXiv:2307.13854*, 2023b.

## A  BASELINE IMPLEMENTATIONS

We follow the existing frameworks to set up baselines in each benchmark. In SWE-bench (Jimenez et al., 2023), we follow the prompt styles of the Agentless (Xia et al., 2024) pipeline to first localize suspicious files, then find classes and functions to edit. In WebArena (Zhou et al., 2023b), we follow the implementation of Step (Sodhi et al., 2024), which concatenates task objectives, action space descriptions, general instructions (e.g., output formats) and webpage observations in the prompt, and ask LMs to predict the next action. By default, we use the accessibility tree[4] as the observation space. In OSWorld (Xie et al., 2024) and Spider2-V (Cao et al., 2024), we follow the original prompt style designed by the benchmark, which also concatenates task objectives, action space descriptions, general instructions and computer observations in the prompt. By default, we use the accessibility tree as the observation space for OSWorld, and use the set-of-mark for Spider2-V due to the significant information loss of the accessibility tree in the original benchmark. See an example in Table 22 and 23 for more details.

## B  COMPARISON TO TASK-SPECIFIC APPROACHES

In Table 6, we compare LEARN-BY-INTERACT to top-10 task-specific approaches (with open-sourced code) that may not broadly applied in agent scenarios for SWE-bench (Zan et al., 2024) and WebArena (Zhou et al., 2023b). All the information is retrieved on 2024-10-01 from the official leaderboard `https://www.swebench.com/` and `https://docs.google.com/spreadsheets/d/1M801lEpBbKSNwP-vDBkC_pF7LdyGU1f_ufZb_NWNBZQ/edit?gid=0#gid=0`. To the best of our knowledge, we are the first to apply our methods in OSWorld (Xie et al., 2024) and Spider2-V (Cao et al., 2024).

## C  DATASET EXAMPLES

From Table 8 to 21, we provide one example for each dataset with full instructions, interaction history with the environment.

## D  EXPERIMENTAL SETTINGS

We retrieve documents until the maximum length of LLMs for RAG and set an upper bound number of 50 documents, where the retrieved documents remain unchanged throughout agent interaction trajectory because only instructions are used as the query for retrieval. For Reflexion (Shinn et al.,

---

[4]`https://developer.mozilla.org/en-US/docs/Glossary/Accessibility_tree`

Table 7: Top-10 results of WebArena from the leaderboard at `https://docs.google.com/spreadsheets/d/1M8O1lEpBbKSNwP-vDBkC_pF7LdyGU1f_ufZb_NWNBZQ/edit?gid=0#gid=0`. All the numbers are fetched on 2024-10-01.

| Approach ↓ | site | result |
|---|---|---|
| Jace.AI | `https://www.jace.ai/` | 57.1 |
| WebPilot | `https://www.arxiv.org/pdf/2408.15978` | 37.2 |
| AWM | `https://arxiv.org/pdf/2409.07429` | 35.5 |
| Step | `https://arxiv.org/abs/2310.03720` | 33.5 |
| BrowserGym | `https://github.com/ServiceNow/BrowserGym` | 23.5 |
| Auto Eval | `https://arxiv.org/abs/2404.06474` | 20.2 |
| Tree Search | `https://jykoh.com/search-agents` | 19.2 |
| AutoWebGLM | `https://arxiv.org/abs/2404.03648` | 18.2 |
| gpt-4-0613 | `https://arxiv.org/abs/2307.13854` | 14.9 |
| gpt-4o-2024-05-13 | `https://arxiv.org/abs/2307.13854` | 13.1 |
| LEARN-BY-INTERACT (ours) | This paper | 39.2 |

2024), we use the maximum trials 3. In LATS (Zhou et al., 2023a), we use the number of generated action 5, depth limit 15, value function weight 0.8, following the original setting in paper with WebShop (Yao et al., 2022a), which is also an agent task based on website. By default, we use https://huggingface.co/dunzhang/stella_en_1.5B_v5 as the retriever for model-based retrieval considering both size and the performance. We use the temperature 0 throughout the experiments to ensure better reproductivity of the experiments. During training, we the batch size 128, learning rate 0.00002, warmup ratio 0.03 and maximum length 8192, and tune the model for 3 epochs. All experiments are conducted in H100 machines with 80GB memeory.

# E  DOCUMENT SOURCES

We use all the non-repeated python files in SWE-bench-Lite (Jimenez et al., 2023) as the document sources. Although we may not always find abundant documentations and tutorials for each environment, we believe that documentations in the same domain still have a good coverage of frequent operations. For example, one subset of WebArena (Zhou et al., 2023b) focuses on the navigation of the shopping website OneStopMarket, we use the Amazon documentation as a good replacement. Regardless of the shopping websites, the frequent tasks usually include order change, product search, delivery checking, etc. Therefore, we use other documentations in the same domain to sample task instructions when the exact version for the target environment is not available. Concretely, we use the following sources for WebArena:

- https://docs.gitlab.com/ee/tutorials/
- https://support.google.com/maps
- https://www.amazon.com/hz/contact-us/foresight/hubgateway
- https://support.reddithelp.com/hc/en-us/articles

The following sources are used for OSWorld:

- https://support.google.com/chrome/?hl=en
- https://www.gimp.org/tutorials/
- https://books.libreoffice.org/en/CG72/CG72.html
- https://books.libreoffice.org/en/WG73/WG73.html
- https://ubuntu.com/tutorials/command-line-for-beginners
- https://support.mozilla.org/en-US/products/thunderbird
- https://wiki.videolan.org/Documentation:Documentation
- https://code.visualstudio.com/docs

, The following sources are used for Spider2-V:

- https://docs.getdbt.com/
- https://release-1-7-2.dagster.dagster-docs.io/
- https://docs.astronomer.io/
- https://docs.airbyte.com/
- https://airbyte.com/tutorials/
- https://airbyte-public-api-docs.s3.us-east-2.amazonaws.com/rapidoc-api-docs.html
- https://superset.apache.org/docs/
- https://www.metabase.com/docs/v0.49/
- https://www.metabase.com/learn/
- https://docs.snowflake.com/en/
- https://cloud.google.com/bigquery/docs/
- https://jupyterlab.readthedocs.io/en/4.1.x/

## F  SYNTHESIZED DATA EXAMPLES

From Table 24 to 30, we provide a complete example of data synthesis. To begin with, an LLM generates instructions based on standard resources like tutorials, documentations and FAQs: Upload CSV data in Google Drive to BigQuery. (See prompt in Table 33) It then attempts solve the task by predicting actions and collecting feedback from environments (interactions). This produces a long trajectory showing how LLMs try to achieve the goal.

However, it is not guaranteed that the trajectory successfully achieves the target. In our example, the LLM makes a wrong prediction in the action 4. It selects the table source Google Cloud Storage, while the correct action should select "Drive" to align with the instruction that reuiqres to upload CSV data in Google Drive. This results in wrong actions in the subsequent predictions, and the generated trajectory is not aligned with the initial instruction, which leads to noisy data in this case.

Instead of using the original instruction-trajectory pairs for downstream training and in-context learning, we fix the mentioned misalignment by crafting new instructions for each sub-trajectory (backward construction). Concretely, we feed the generated trajectory into LLM prompts, and ask it to summarize the trajectory or propose a new task based on it. For example, the LLM updates the task objective to "Link CSV file in Google Cloud Storage to BigQuery" after observing the trajectory, which makes the task instrucion and the trajectory aligned. Additionally, we also generate new instructions for each sub-trajectory, which would increase the utility of a generated full trajectory. For instance, based on the sub-trajectory (observation 0, Action 1, observation 1), the LLM generates a new instruction: When is dataset "demo" created? In Table 31 and 32, we list more generated instructions based on sub-trajectories.

## G  CASE STUDY ON FILTERED EXAMPLES

In Table 40-49, we demonstrate the representative synthesized examples that fail to meet our designed criteria. The example in Table 40-45 is filtered because the trajectory shows detour in accomplishing the goal, i.e. Action 1-6 are not necessary. The example in Table 46-49 is filtered because it goes back and forth in states, i.e. repeat the actions of clicking "My Orders" and clicking "View Order". We filter these low-quality examples to avoid their negative influences in the downstream applications.

Table 8: **SWE-bench example**

| **Instruction** |
| --- |

Please address the following Github issue.

### GitHub Problem Description ###
ModelChoiceIteratorValue is not hashable.
Description

Recently I migrated from Django 3.0 to Django 3.1. In my code, I add custom data-* attributes to the select widget options. After the upgrade some of those options broke. Error is TypeErrorunhashable type: 'ModelChoiceIteratorValue'.
Example (this one breaks):
```
    def create_option(self, name, value, label, selected, index, subindex=None, attrs=None):
        context = super().create_option(name, value, label, selected, index, subindex, attrs)
        if not value:
            return context
        if value in self.show_fields: # This is a dict 1: ['first_name', 'last_name']
            context['attrs']['data-fields'] = json.dumps(self.show_fields[value])
```
However, working with arrays is not an issue:
```
    def create_option(self, name, value, label, selected, index, subindex=None, attrs=None):
        context = super().create_option(name, value, label, selected, index, subindex, attrs)
        if not value:
            return context
        if value in allowed_values: # This is an array [1, 2]
            ...
```

| **Observation 0 (repository structure)** |
| --- |

```
django/
    setup.py
    __init__.py
    __main__.py
    shortcuts.py
    forms/
        __init__.py
        utils.py
        fields.py
        renderers.py
        forms.py
        formsets.py
        widgets.py
        boundfield.py
        models.py
    db/
        __init__.py
        transaction.py
        utils.py
        models/
            signals.py
            constants.py
            fields/
                files.py
                __init__.py
                related.py
...
```

Table 9: **SWE-bench example cont.**

**Action 1 (Find related files to edit)**

Based on the problem description and the repository structure, here are the most likely files that would need to be edited to fix the issue:

django/forms/models.py
django/forms/fields.py
django/forms/widgets.py
django/forms/__init__.py
django/db/models/fields/__init__.py
django/db/models/fields/related.py
...

**Observation 1 (File skeleton)**

### File: django/forms/models.py
def construct_instance(form, instance, fields=None, exclude=None):
    ...
def model_to_dict(instance, fields=None, exclude=None):
    ...
class ModelChoiceIteratorValue:
    def __init__(self, value, instance):
        ...
    def __str__(self):
        ...
    def __eq__(self, other):
        ...
class ModelChoiceField(ChoiceField):
    def _get_choices(self):
        ...
### File: django/forms/fields.py
class Field:
    widget = TextInput # Default widget to use when rendering this type of Field.
    hidden_widget = HiddenInput # Default widget to use when rendering this as "hidden"

    def __init__(self, *, required=True, widget=None, label=None, initial=None):
    ...
    def prepare_value(self, value):
    ...
    def to_python(self, value):
    ...
    def validate(self, value):
    ...
    def run_validators(self, value):
    ...
class ChoiceField(Field):
    def validate(self, value):
        ...
    def valid_value(self, value):
        ...
...

Table 10: **SWE-bench example cont.**

| **Action 2 (Find related classes, methods to edit)** |
| --- |

Here are the identified locations that need inspection or editing to fix the problem:

django/forms/models.py
class: ModelChoiceIteratorValue

django/forms/models.py
class: ModelChoiceField
method: _get_choices

django/forms/fields.py
class: ChoiceField
method: valid_value
...

| **Observation 2 (Classes and methods content)** |
| --- |

```
### File django/forms/models.py
class ModelChoiceIteratorValue:
    def __init__(self, value, instance):
        self.value = value
        self.instance = instance

    def __str__(self):
        return str(self.value)

    def __eq__(self, other):
        if isinstance(other, ModelChoiceIteratorValue):
            other = other.value
        return self.value == other

class ModelChoiceField(ChoiceField):

    def _get_choices(self):
        return self.iterator(self)

### django/forms/fields.py
class ChoiceField(Field):

    def valid_value(self, value):
        text_value = str(value)
        for k, v in self.choices:
            if isinstance(v, (list, tuple)):
                # This is an optgroup, so look inside the group for options
                for k2, v2 in v:
                    if value == k2 or text_value == str(k2):
                        return True
            else:
                if value == k or text_value == str(k):
                    return True
        return False
...
```

Table 11: **SWE-bench example cont.**

**Action 3 (Edit code to address the issue)**

Based on the issue description and the provided code segment, it appears that the 'Model-ChoiceIteratorValue' class is not hashable, which is causing the TypeError when trying to use it as a dictionary key. To fix this, we need to make the 'ModelChoiceIteratorValue' class hashable by implementing the '__hash__' method. Here's the proposed fix:

### django/forms/models.py
### Replace the following code
class ModelChoiceIteratorValue:
    def __init__(self, value, instance):
        self.value = value
        self.instance = instance

    def __str__(self):
        return str(self.value)

    def __eq__(self, other):
        if isinstance(other, ModelChoiceIteratorValue):
            other = other.value
        return self.value == other

### with

class ModelChoiceIteratorValue:
    def __init__(self, value, instance):
        self.value = value
        self.instance = instance

    def __str__(self):
        return str(self.value)

    def __eq__(self, other):
        if isinstance(other, ModelChoiceIteratorValue):
            other = other.value
        return self.value == other

    def __hash__(self):
        return hash(self.value)

**Observation 3 (Generated patch)**

diff –git a/django/forms/models.py b/django/forms/models.py
index 5a30801..55b3c81 100644
- - - a/django/forms/models.py
+++ b/django/forms/models.py
@@ -1171,6 +1171,9 @@ class ModelChoiceIteratorValue:
        other = other.value
    return self.value == other

+ def __hash__(self):
+ return hash(self.value)
+

class ModelChoiceIterator:
    def __init__(self, field):

Table 12: **Webarena example**

| **Instruction** |
| --- |
| Tell me the total cost of my latest cancelled order? |

**Observation 0**

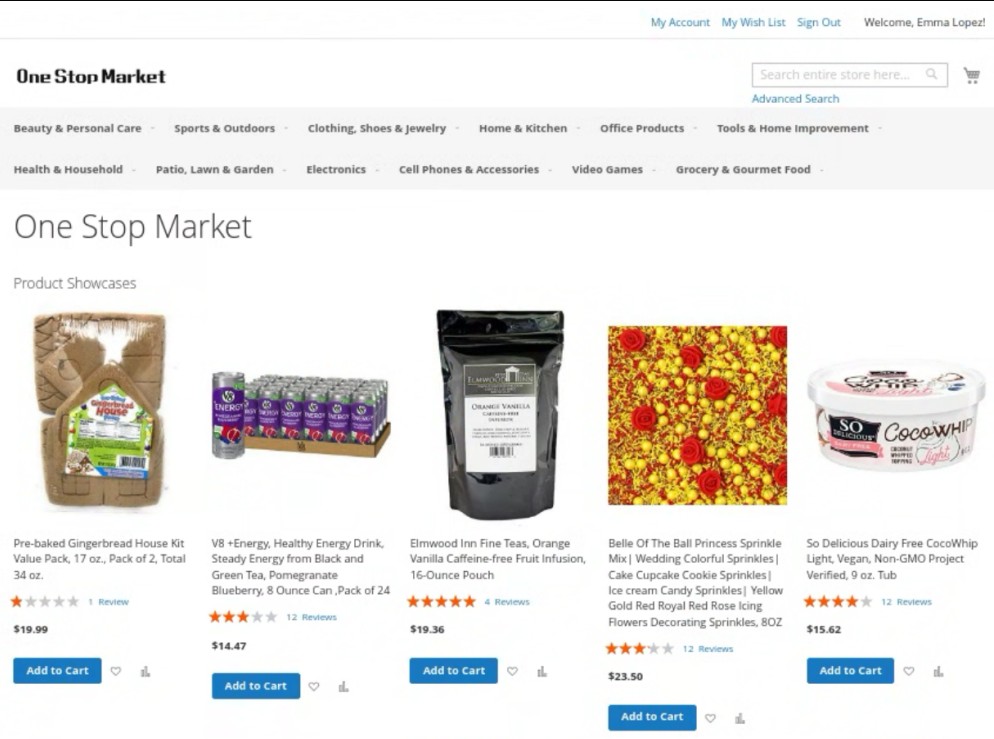

**Action 1**

click [4918] // click the button 'My Account'

Table 13: **Webarena example cont.**

**Observation 1**

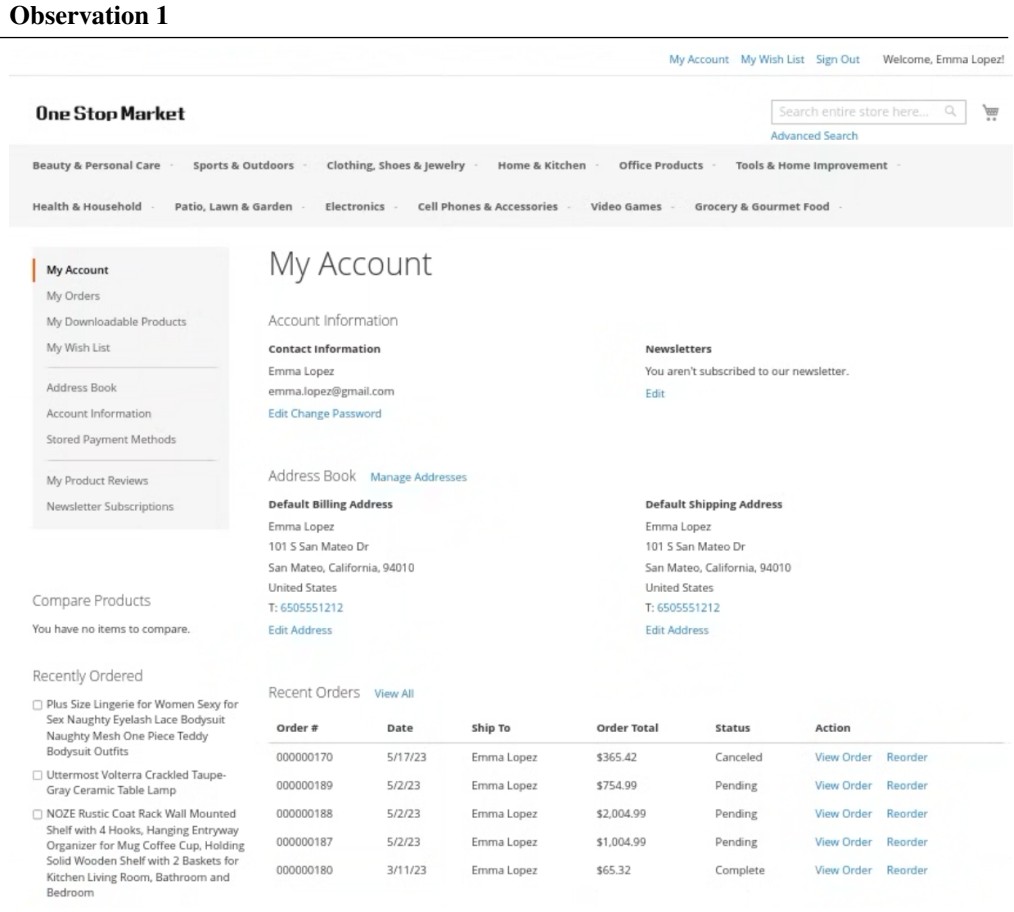

**Action 2**

Stop: The total cost of the latest cancelled order is $365.42

Table 14: **OSWorld example**

---

**Instruction**

---

Could you assist me in adding a new layer and naming it 'Square'?

---

**Observation 0 (Interface of the software GIMP)**

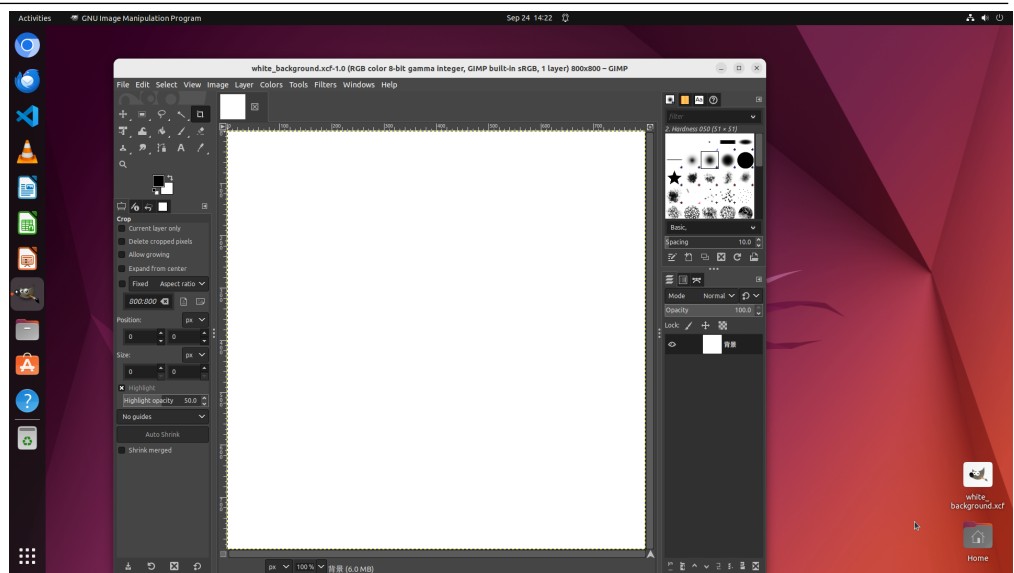

---

**Action 1**

---

import pyautogui
pyautogui.hotkey('shift', 'ctrl', 'n') // shortcut to initialize a new layer.

---

**Observation 1 (Interface of the software GIMP)**

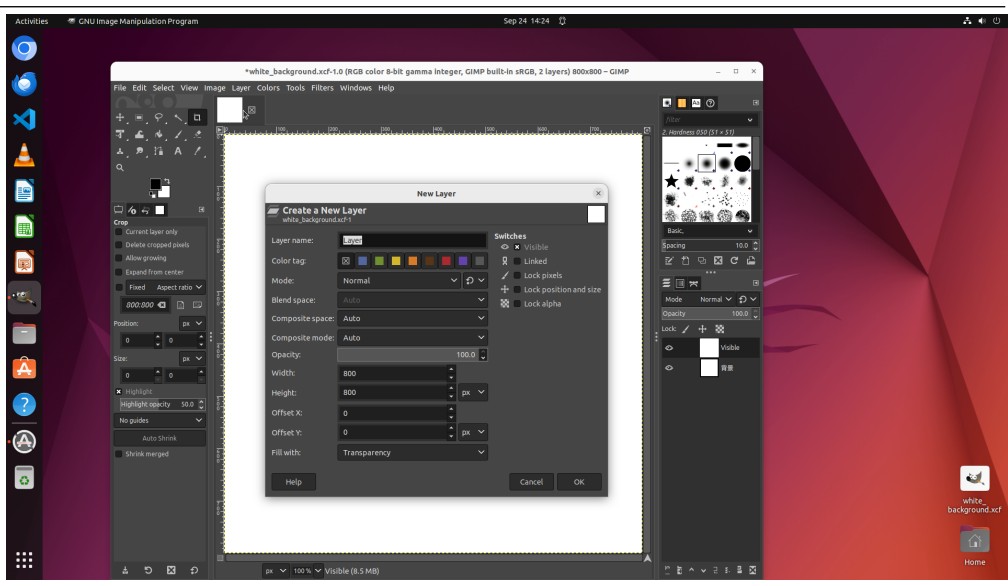

---

Table 15: **OSWorld example cont.**

| **Action 2** |
| --- |

import pyautogui
pyautogui.typewrite('Square') // change the layer name to Square.

**Observation 2 (Interface of the software GIMP)**

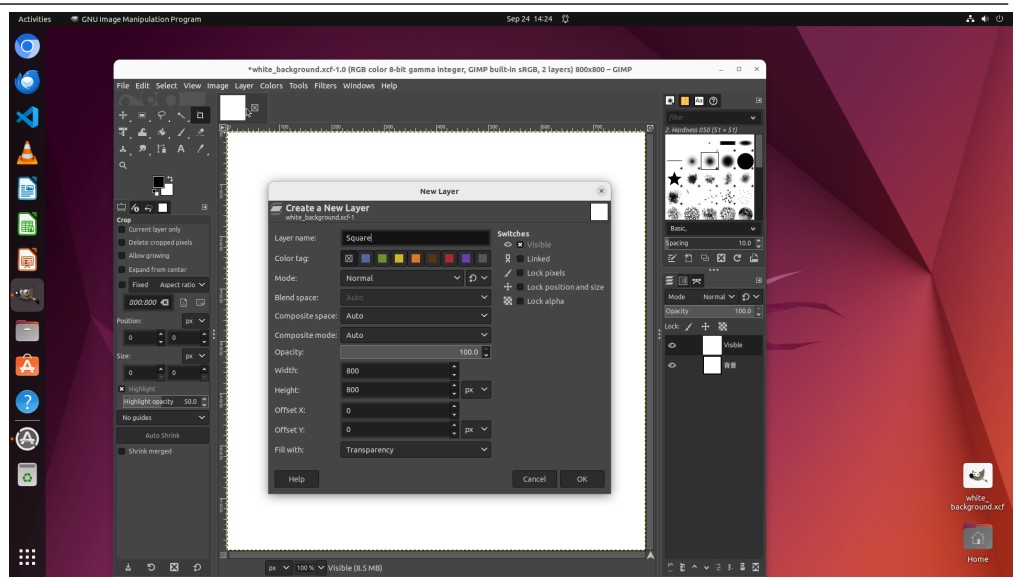

| **Action 3** |
| --- |

import pyautogui
pyautogui.click(1096, 892) // click OK to save the change.

**Observation 3 (Interface of the software GIMP)**

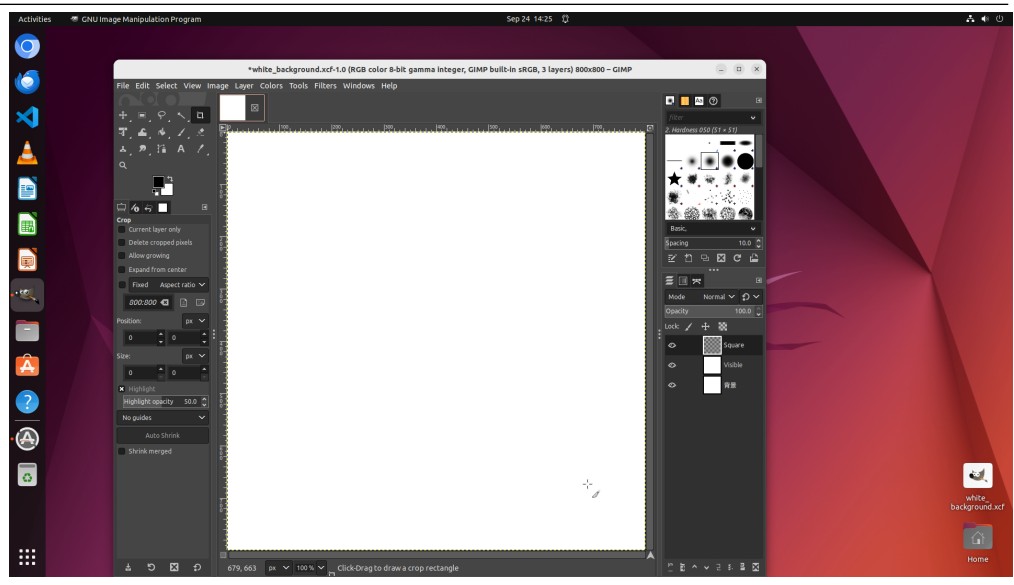

Table 16: **Spider2-V example, cont.**

**Instruction**

I have established a connection from Faker to local .csv file. Could you help me change the running schedule? I hope it can be replicated at 18:00 pm every day.

**Observation 0 (Interface of the software Airbyte)**

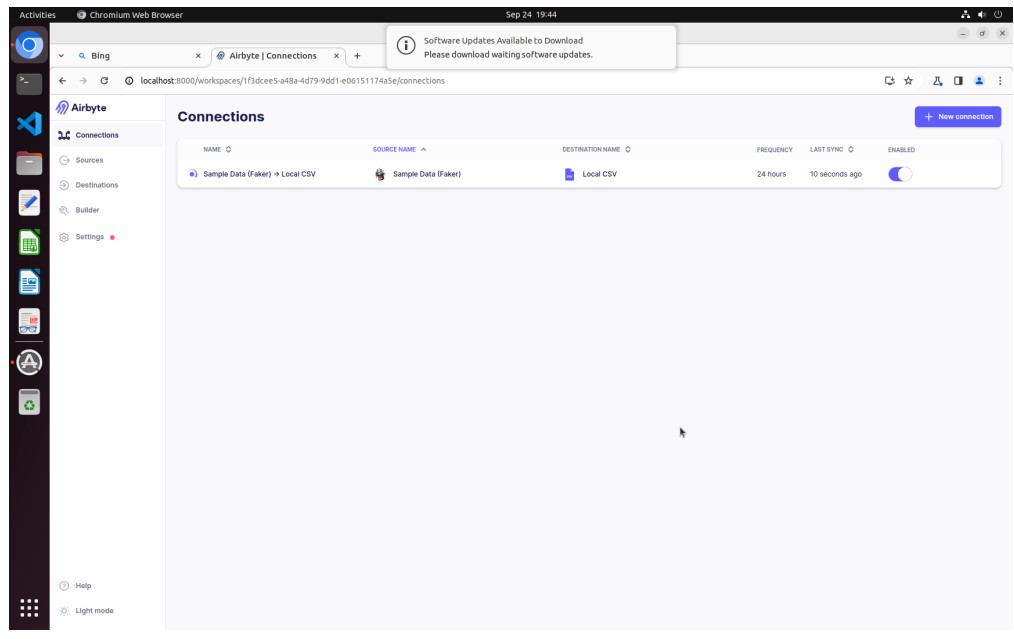

**Action 1**

import pyautogui
pyautogui.click(550,280) // click the connection row with the name "Sample Data (Faker) → Local CSV"

**Observation 1 (Interface of the software Airbyte)**

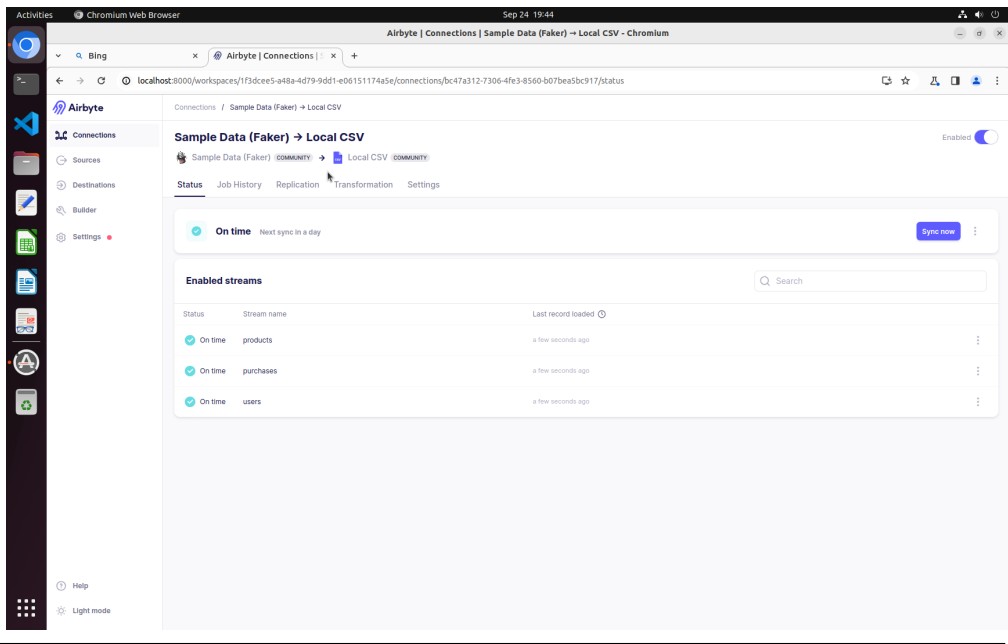

Table 17: **Spider2-V example, cont.**

---

**Action 2**

---

import pyautogui
pyautogui.click(500,300) // click the button Replication

---

**Observation 2 (Interface of the software Airbyte)**

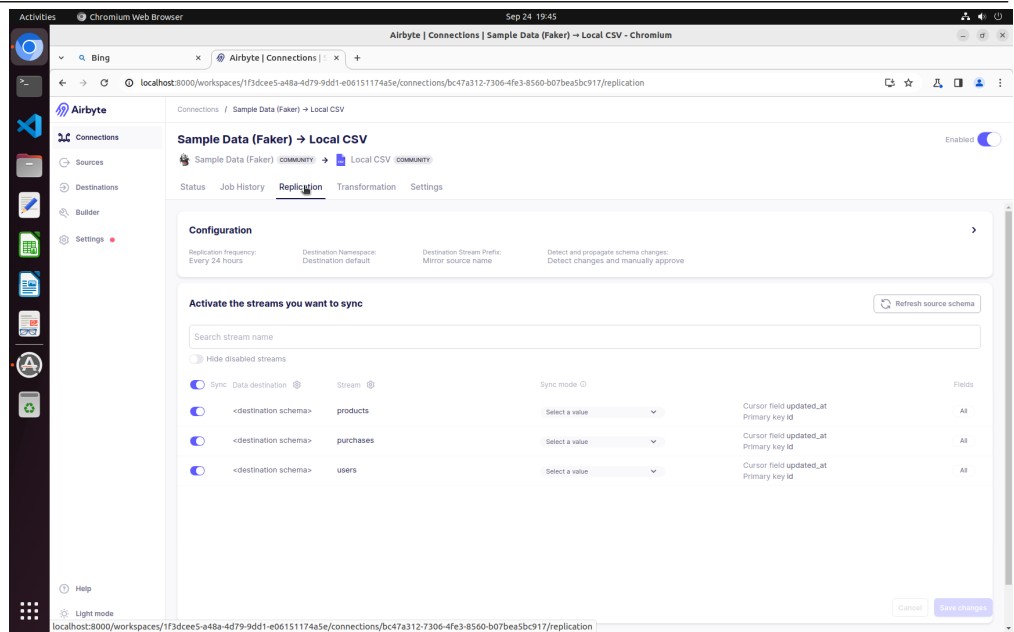

---

**Action 3**

---

import pyautogui
pyautogui.click(500,450) // click the panel Configuration

---

**Observation 3 (Interface of the software Airbyte)**

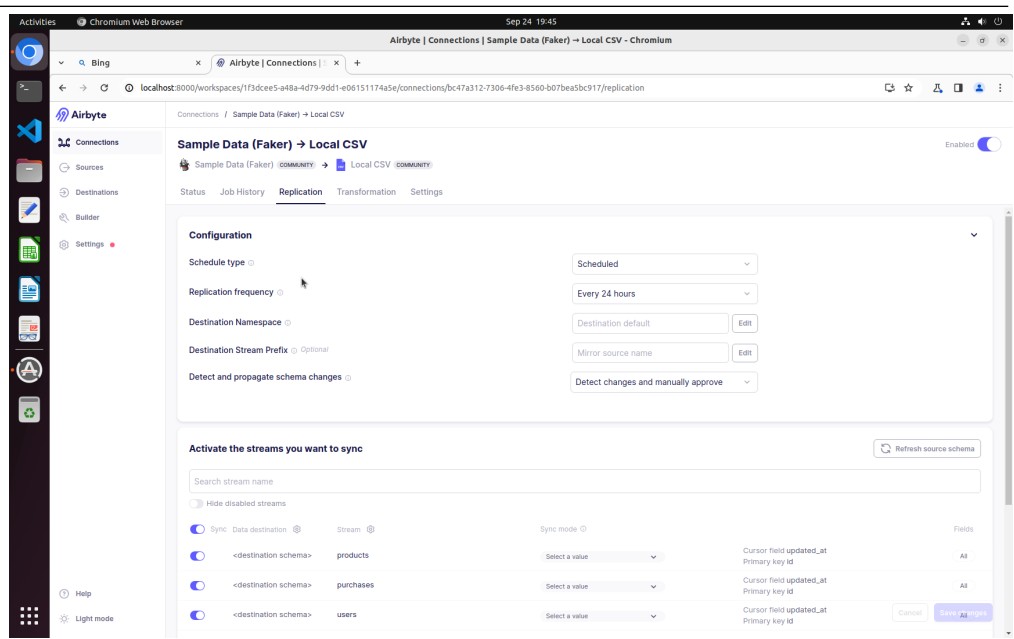

Table 18: **Spider2-V example, cont.**

---

**Action 4**

---

import pyautogui
pyautogui.click(1270,430) // reveal the dropdown menu of the schedule type

---

**Observation 4 (Interface of the software Airbyte)**

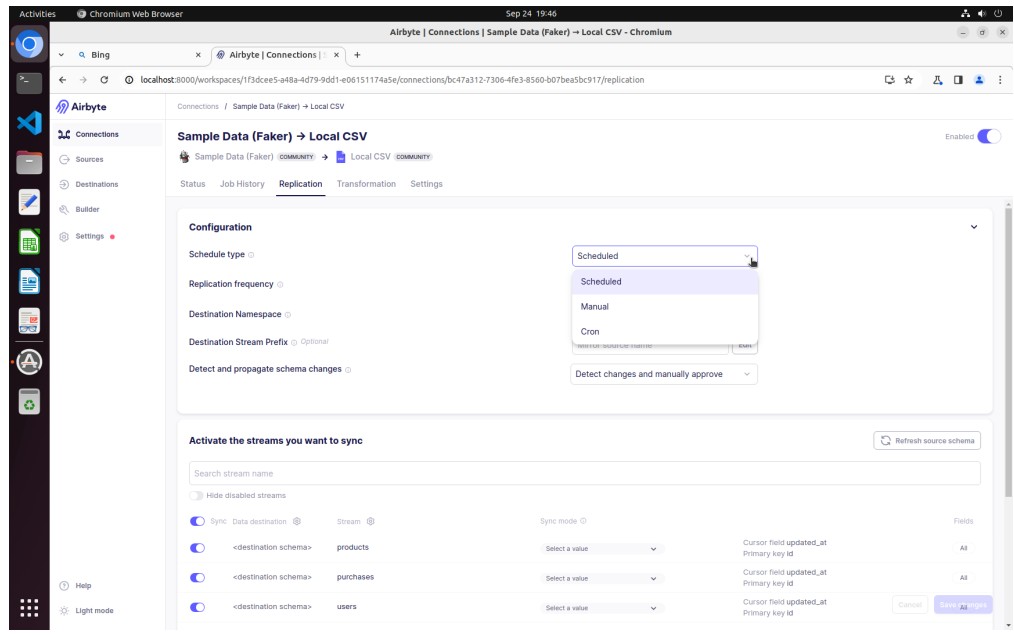

---

**Action 5**

---

import pyautogui
pyautogui.click(1200,565) // select the schedule type Cron

---

**Observation 5 (Interface of the software Airbyte)**

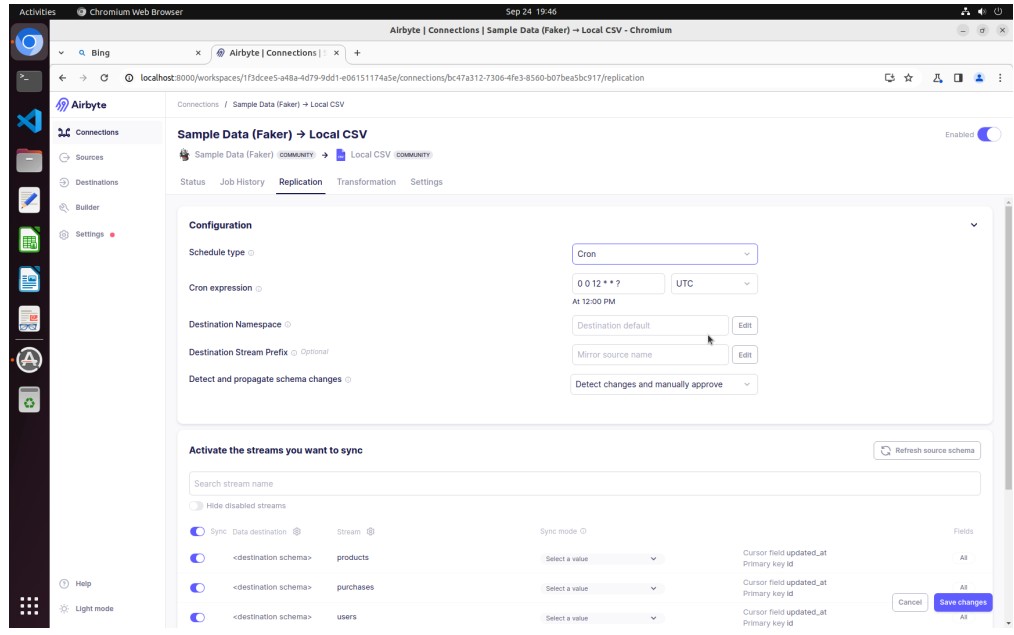

---

Table 19: **Spider2-V example, cont.**

| Action 6 |
| --- |
| import pyautogui
pyautogui.click(1020,480) // focus on the content of the Cron expression |

**Observation 6 (Interface of the software Airbyte)**

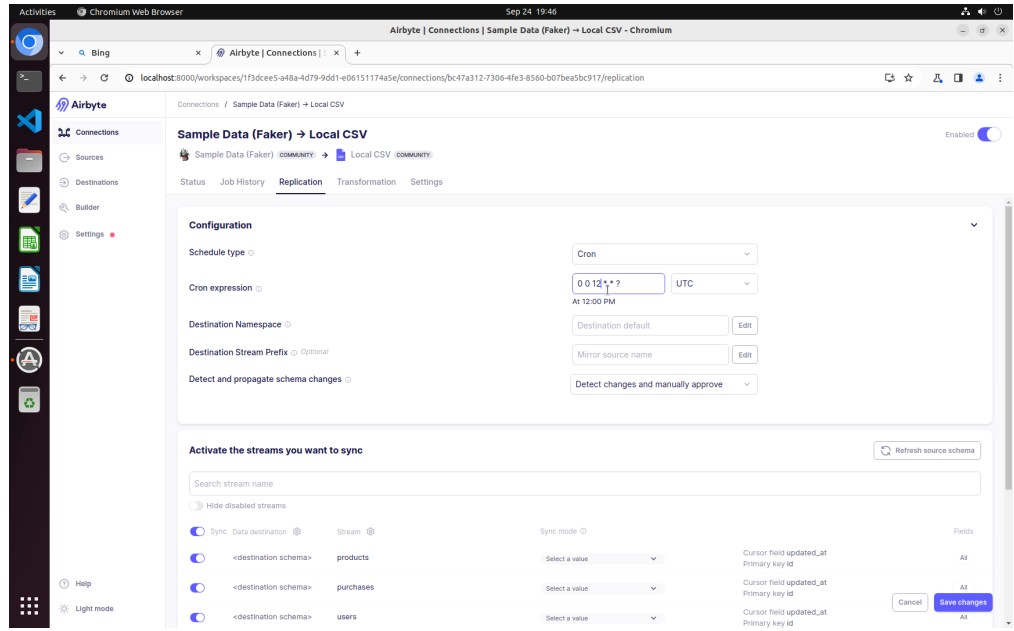

| Action 7 |
| --- |
| import pyautogui
pyautogui.hotkey("ctrl","a") // select all content in the textbox |

**Observation 7 (Interface of the software Airbyte)**

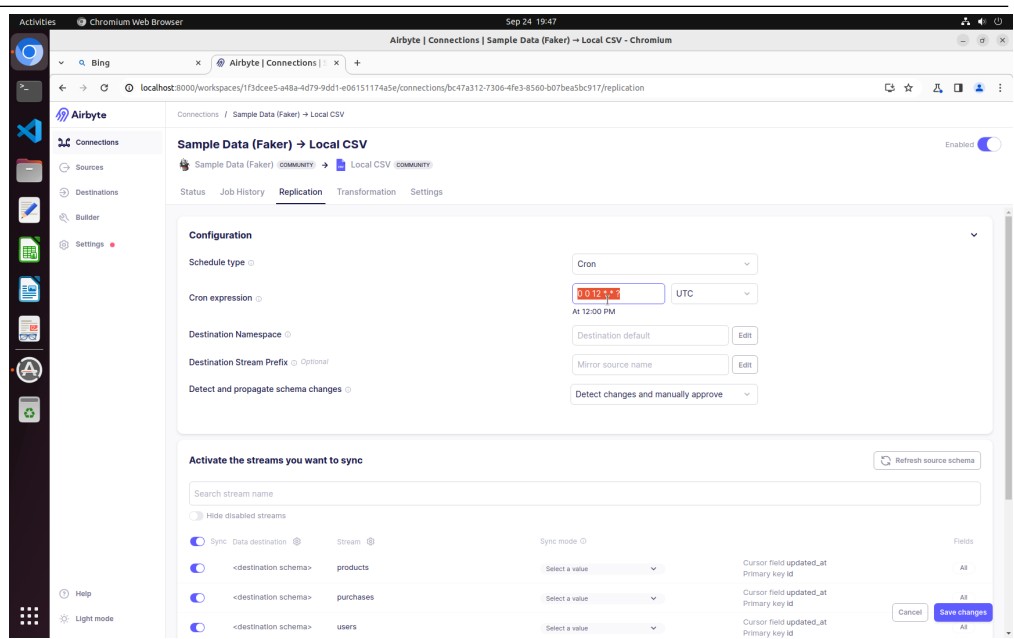

Table 20: **Spider2-V example, cont.**

| **Action 8** |
| --- |

import pyautogui
pyautogui.press(“delete") // delete the default content

| **Observation 8 (Interface of the software Airbyte)** |
| --- |

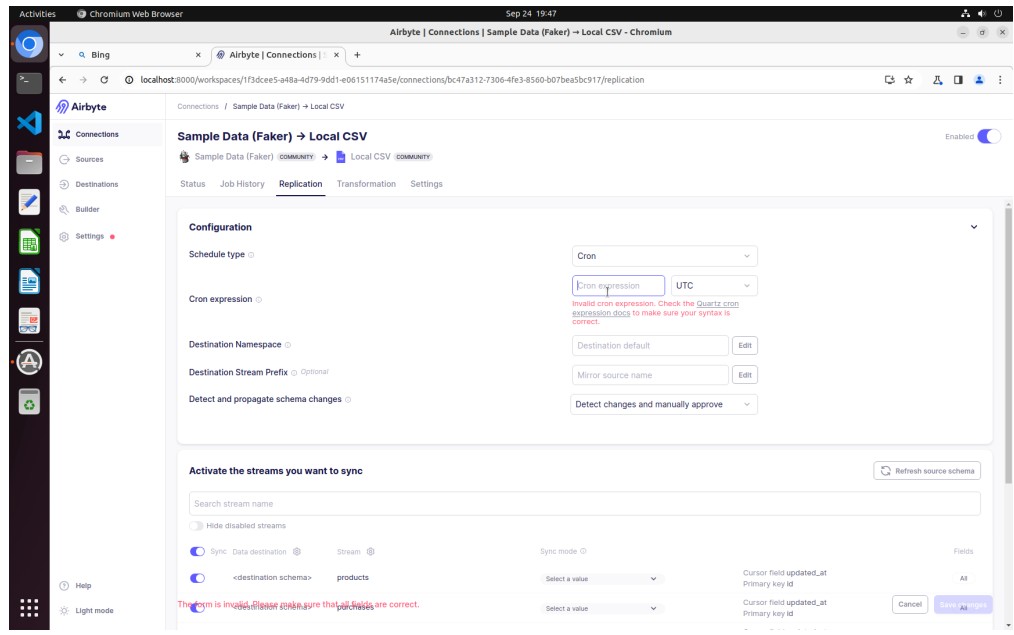

| **Action 9** |
| --- |

import pyautogui
pyautogui.write(“0 0 18 * * ?") // update the content to align 18:00 pm in the instruction

| **Observation 9 (Interface of the software Airbyte)** |
| --- |

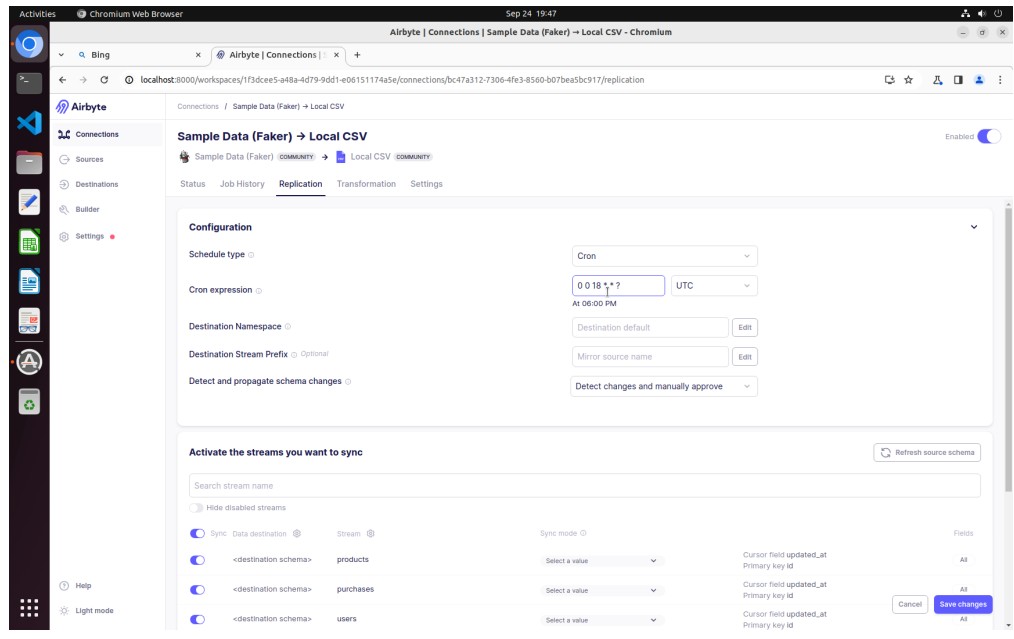

Table 21: **Spider2-V example, cont.**

| **Action 10** |
| --- |
| import pyautogui |
| pyautogui.click(1450,900) // click the button save changes |

**Observation 10 (Interface of the software Airbyte)**

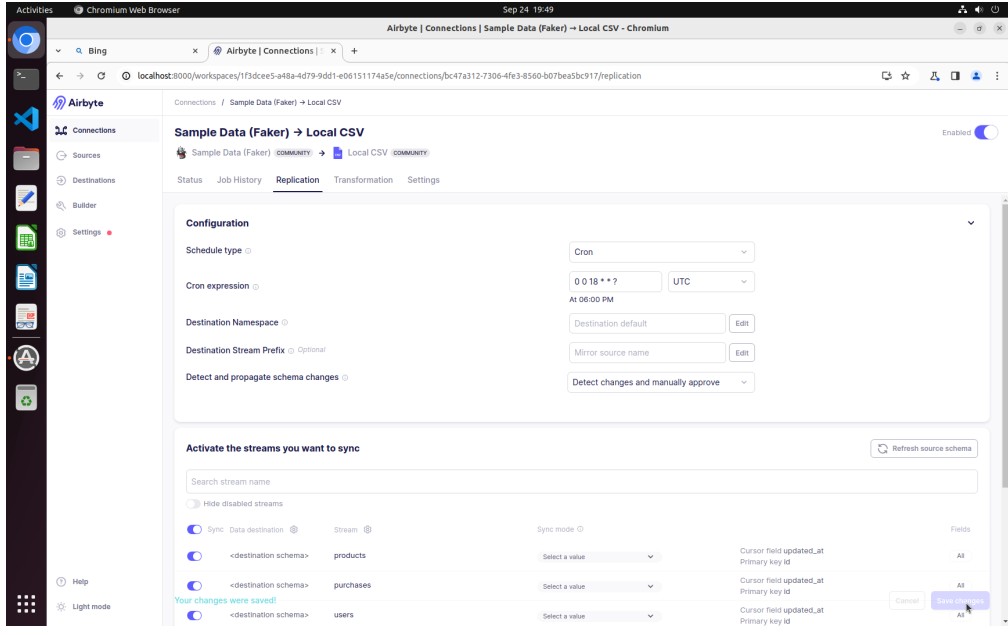

Table 22: Observation space of Spider2-V.

**Screenshot**

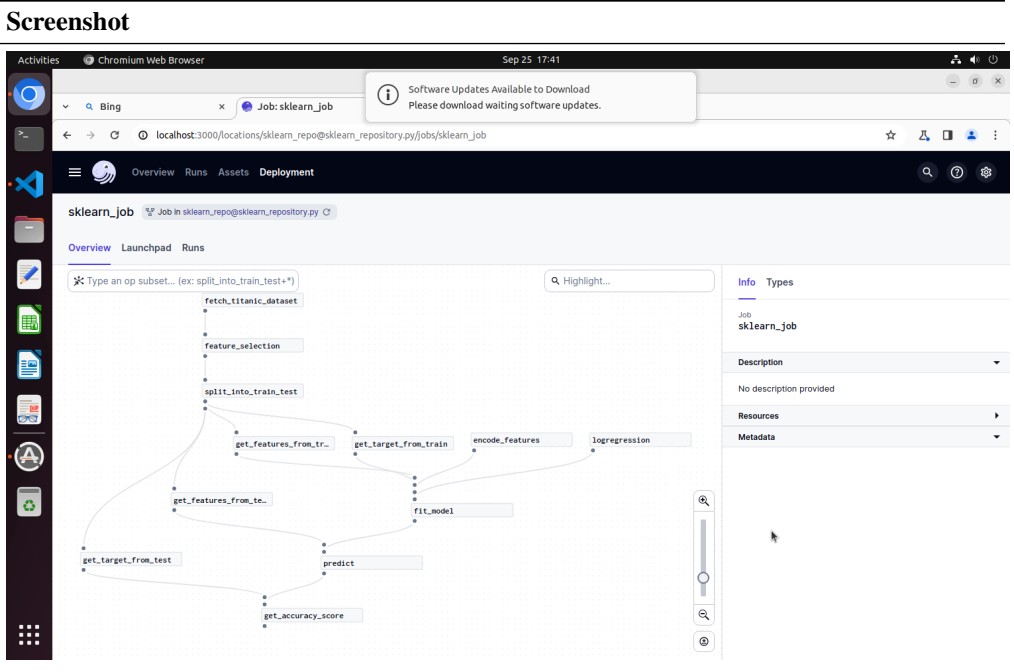

**Set-of-mark**

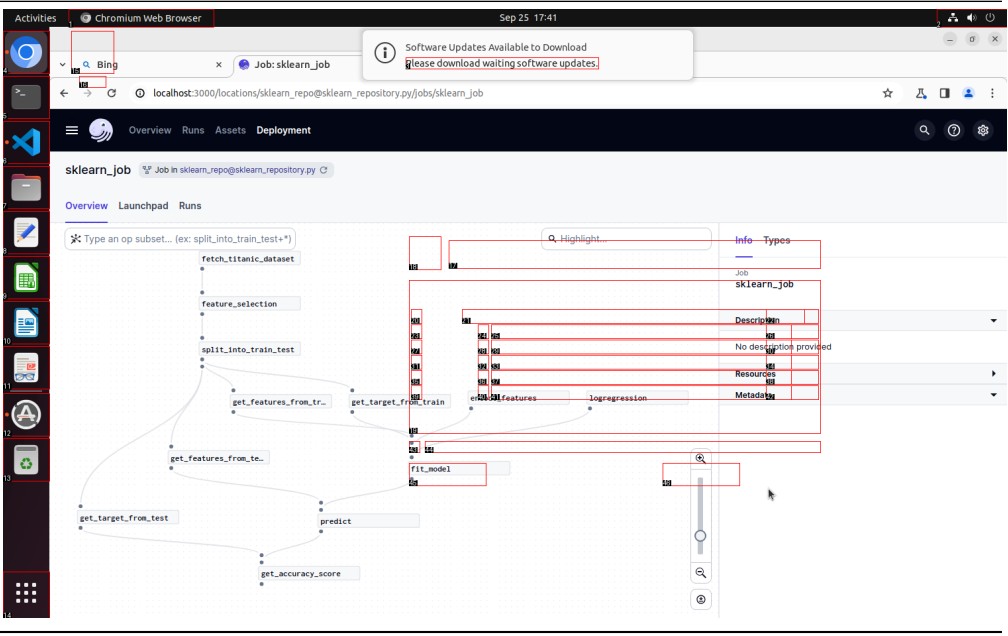

Table 23: Observation space of Spider2-V. The accessibility tree suffers from significant information loss. Compared to the screenshot and set-of-mark shown in Table 22, the presented accessibility tree fails to retrieve webpage information, and only shows the details of the desktop icons in the left panel.

[208, 13] menu Chromium Web Browser ""
[1463, 13] menu System ""
[35, 65] push-button Chromium Web Browser ""
[753, 81] label Please download waiting software updates. ""
[135, 109] label Home
[35, 133] push-button Terminal ""
[35, 201] push-button Visual Studio Code ""
[35, 269] push-button Files ""
[35, 337] push-button Text Editor ""
[953, 370] label Updated software is available for this computer. Do you want to install it now?
[35, 405] push-button LibreOffice Calc ""
[951, 463] table-cell Security updates
[1191, 463] table-cell 638.8 MB
[35, 473] push-button LibreOffice Writer ""
[963, 486] table-cell LibreOffice
[1191, 486] table-cell 23.4 MB
[963, 509] table-cell LibreOffice Calc
[1191, 509] table-cell 8.7 MB
[923, 524] toggle-button Details of updates ""
[963, 532] table-cell LibreOffice Draw
[1191, 532] table-cell 3.0 MB
[35, 541] push-button Document Viewer ""
[963, 555] table-cell LibreOffice Impress
[1191, 555] table-cell 1.3 MB
[963, 578] table-cell LibreOffice Math
[1191, 578] table-cell 673 kB
[35, 612] push-button Software Updater ""
[935, 660] label 1157.8 MB will be downloaded.
[35, 680] push-button Trash ""
[671, 702] push-button Settings. . . ""
[1054, 702] push-button Cancel ""
[1176, 702] push-button Install Now ""
[35, 884] toggle-button Show Applications ""

Table 24: **Example of data synthesis - Bigquery**

| Instruction |
| --- |
| Upload CSV data in Google Drive to BigQuery. |

**Observation 0 (Bigquery Interface)**

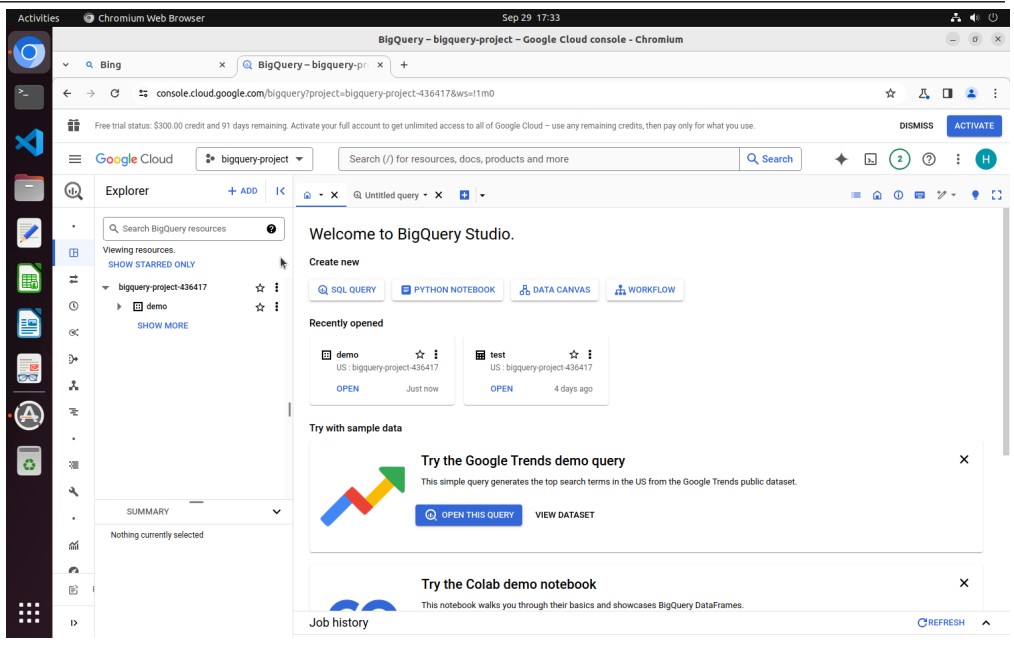

**Action 1**

import pyautogui
pyautogui.doubleClick(332,447) // double click the dataset demo.

**Observation 1 (Bigquery Interface)**

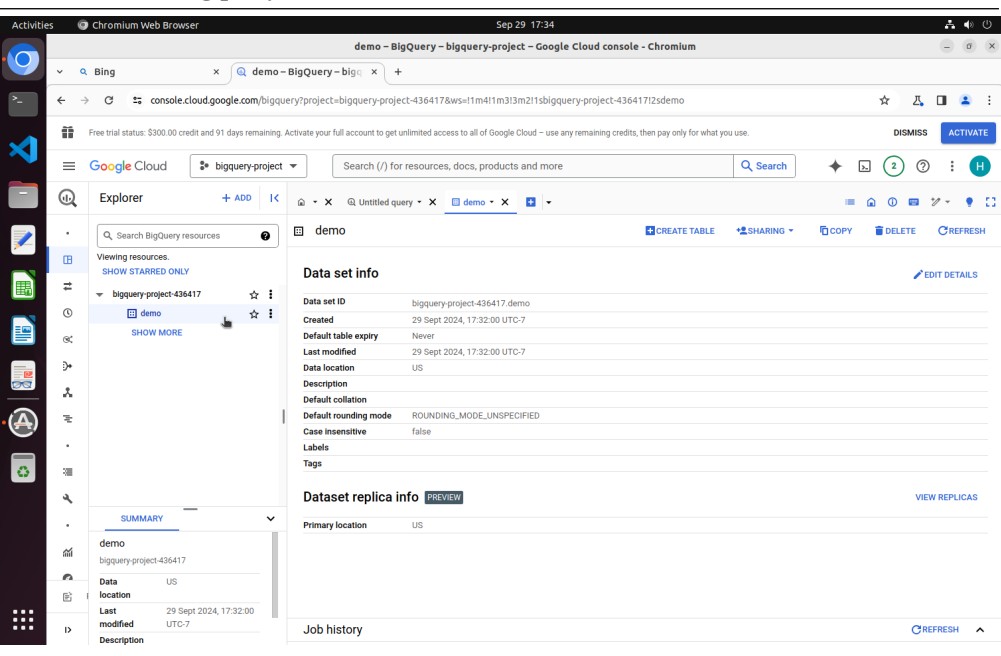

Table 25: **Example of data synthesis - Bigquery**

| **Action 2** |
| --- |

import pyautogui
pyautogui.doubleClick(1002,321) // double click the button CREATE TABLE.

**Observation 2 (Bigquery Interface)**

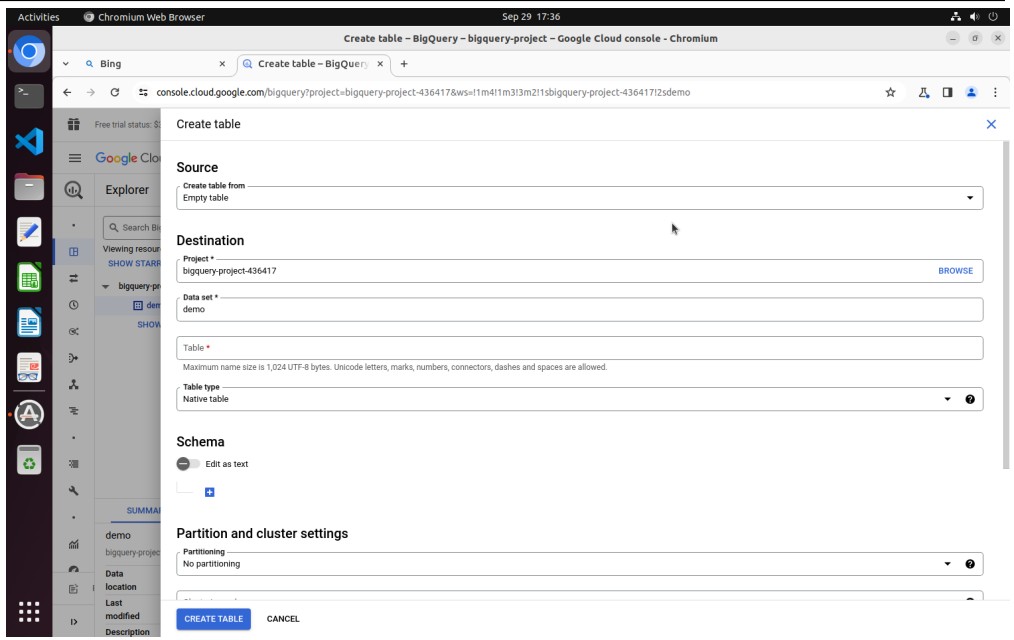

| **Action 3** |
| --- |

import pyautogui
pyautogui.click(1458,279) // click the dropdown menu to select table source.

**Observation 3 (Bigquery Interface)**

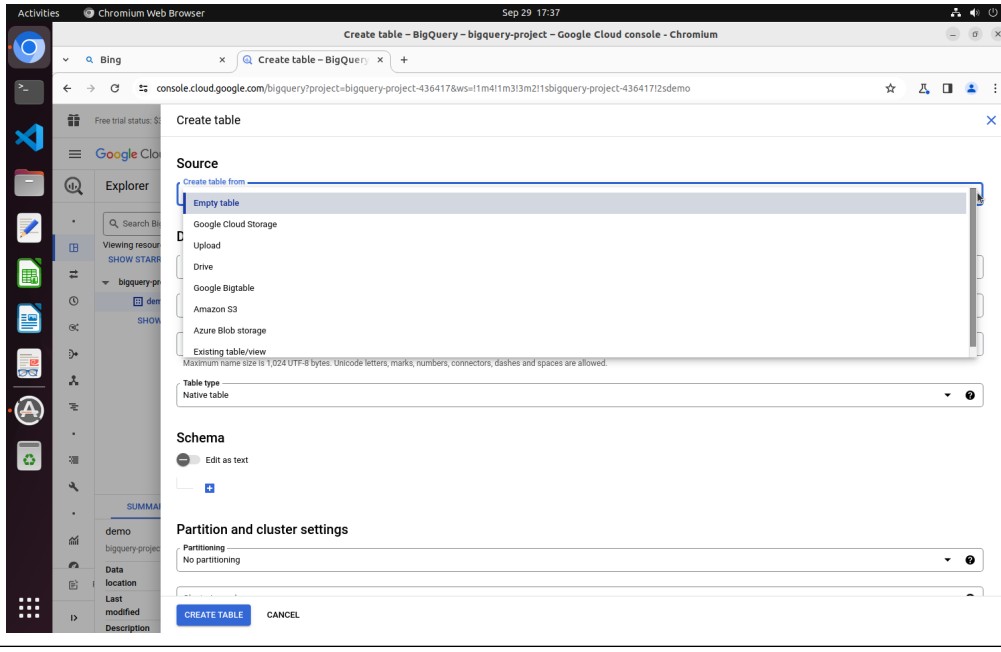

Table 26: **Example of data synthesis - Bigquery**

| **Action 4** |
| --- |
| import pyautogui
// wrong prediction: click the table source Google Cloud Storage.
// The correct prediction should select Drive to align with the instruction.
pyautogui.click(1302,331) |
| **Observation 4 (Bigquery Interface)** |

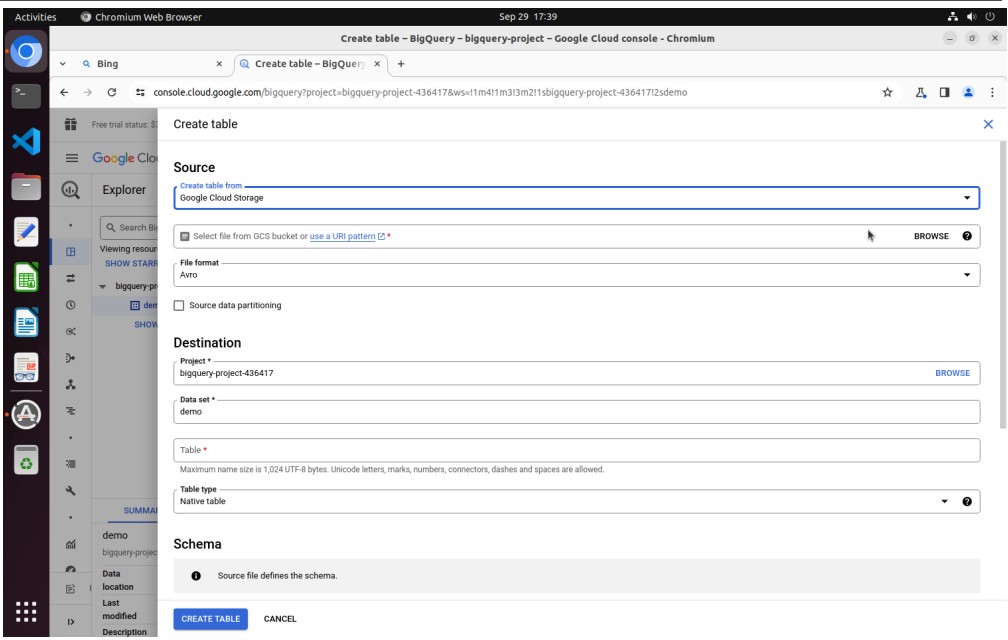

| **Action 5** |
| --- |
| import pyautogui
pyautogui.click(1389,340) // click BROWSE to find file. |
| **Observation 5 (Bigquery Interface)** |

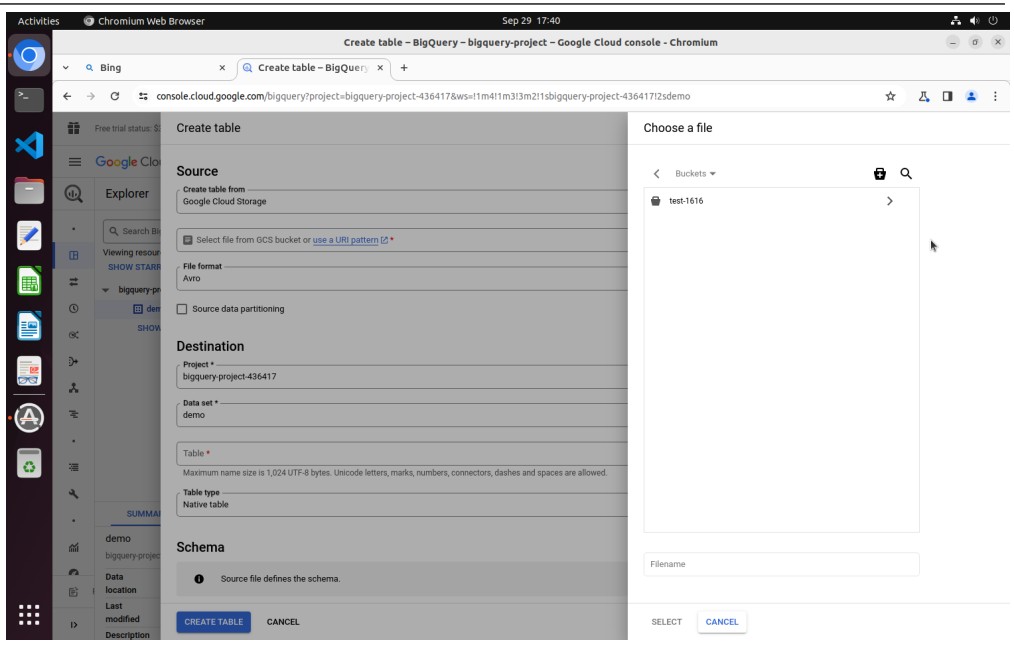

Table 27: **Example of data synthesis - Bigquery**

| Action 6 |
| --- |

import pyautogui
pyautogui.click(1341,282) // click to find files under directory.

**Observation 6 (Bigquery Interface)**

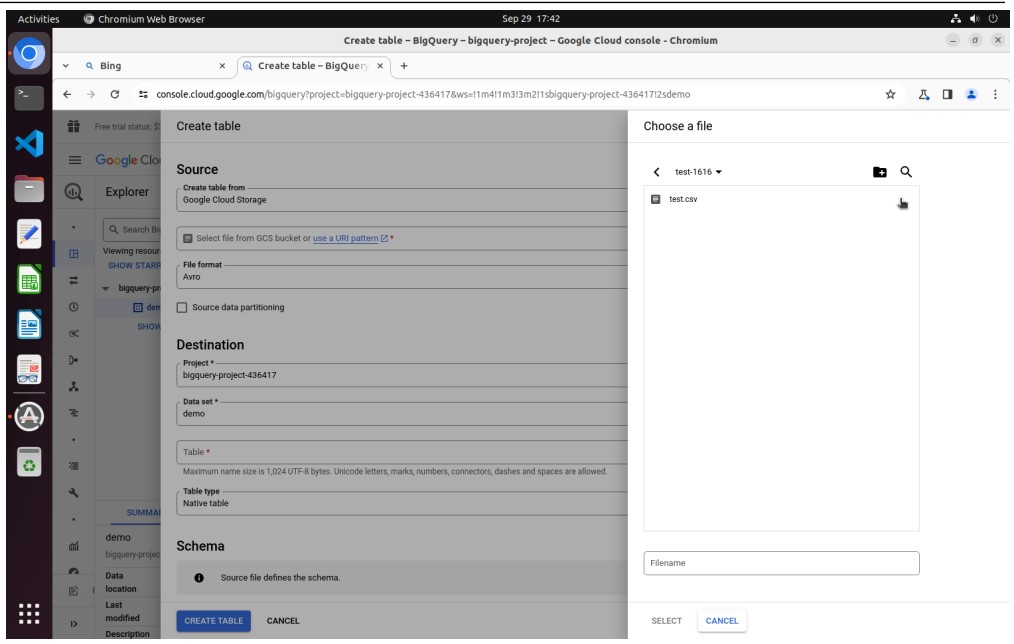

| Action 7 |
| --- |

import pyautogui
pyautogui.doubleClick(1203,284) // select the CSV file.

**Observation 7 (Bigquery Interface)**

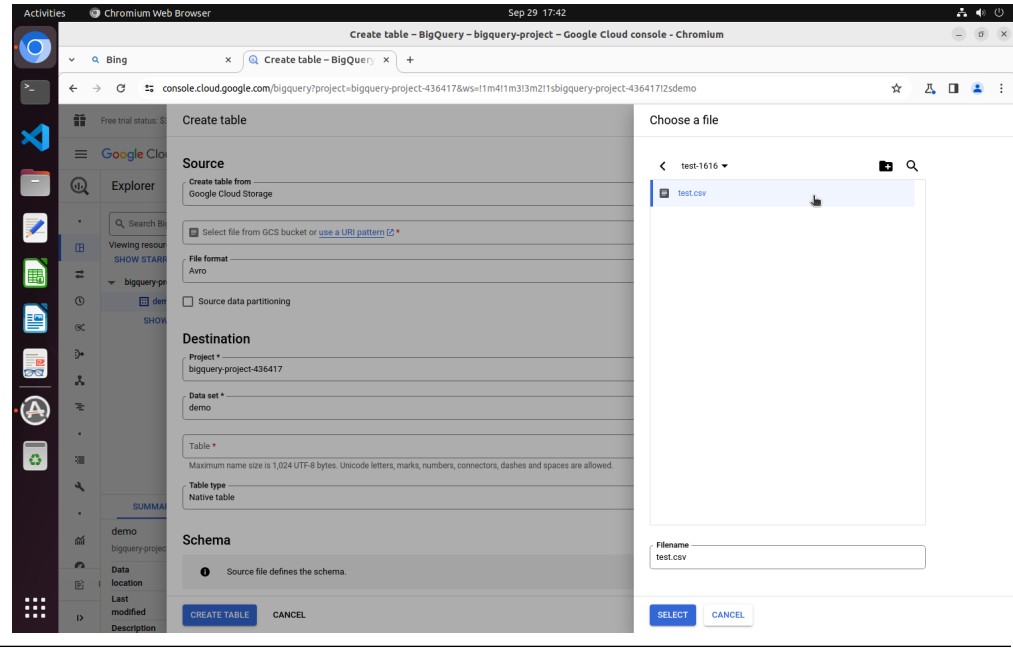

Table 28: **Example of data synthesis - Bigquery**

---

**Action 8**

---

import pyautogui
pyautogui.click(1000,915) // click to select file.

---

**Observation 8 (Bigquery Interface)**

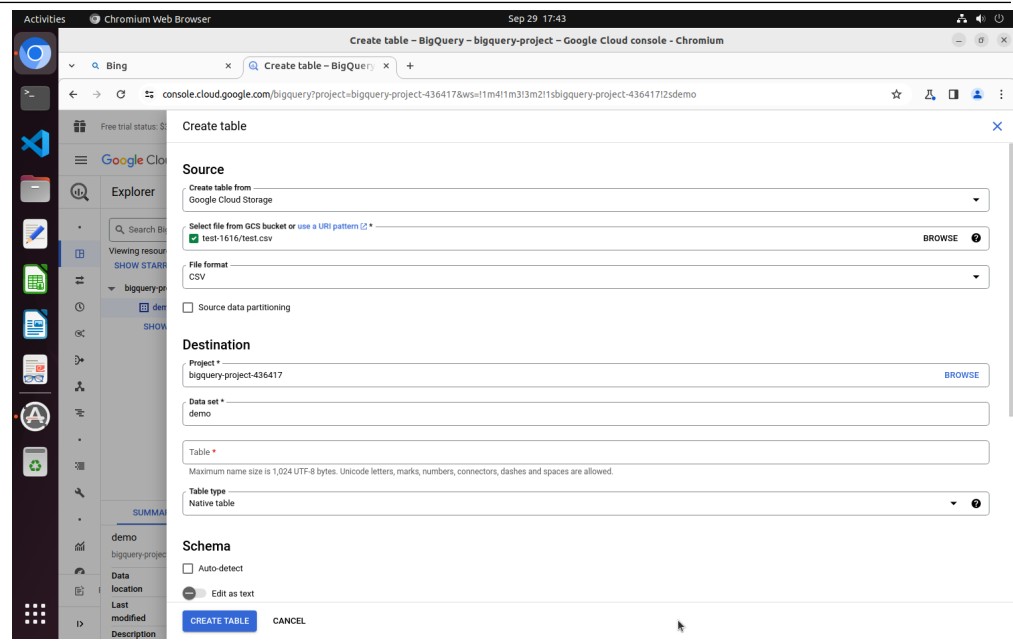

---

**Action 9**

---

import pyautogui
pyautogui.click(1000,668) // click to focus on the text box Table.

---

**Observation 9 (Bigquery Interface)**

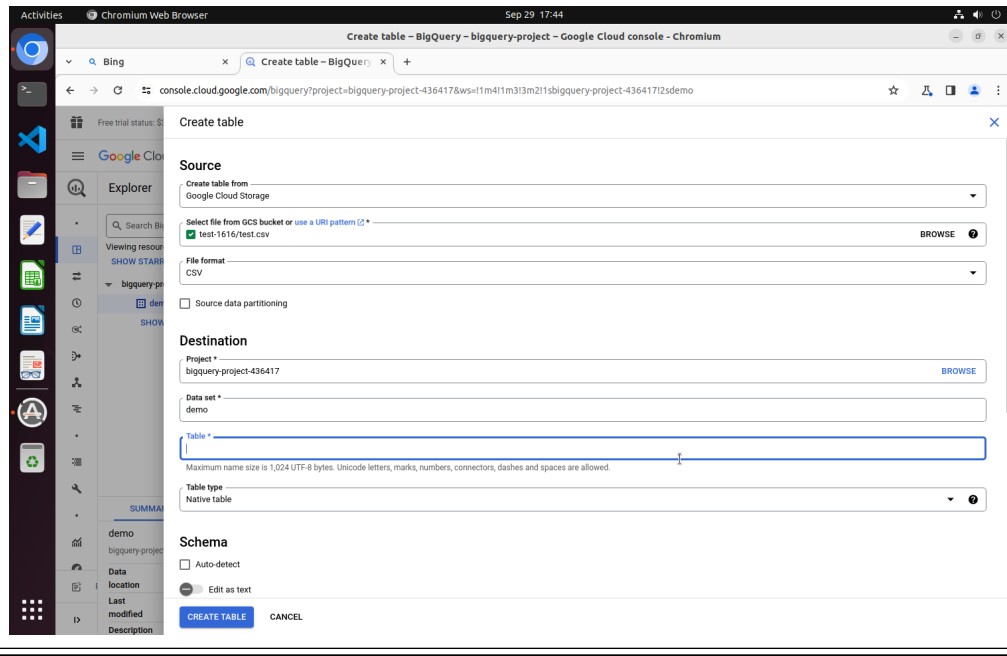

---

Table 29: **Example of data synthesis - Bigquery**

**Action 10**

import pyautogui
pyautogui.write("test") // name the file "test".

**Observation 10 (Bigquery Interface)**

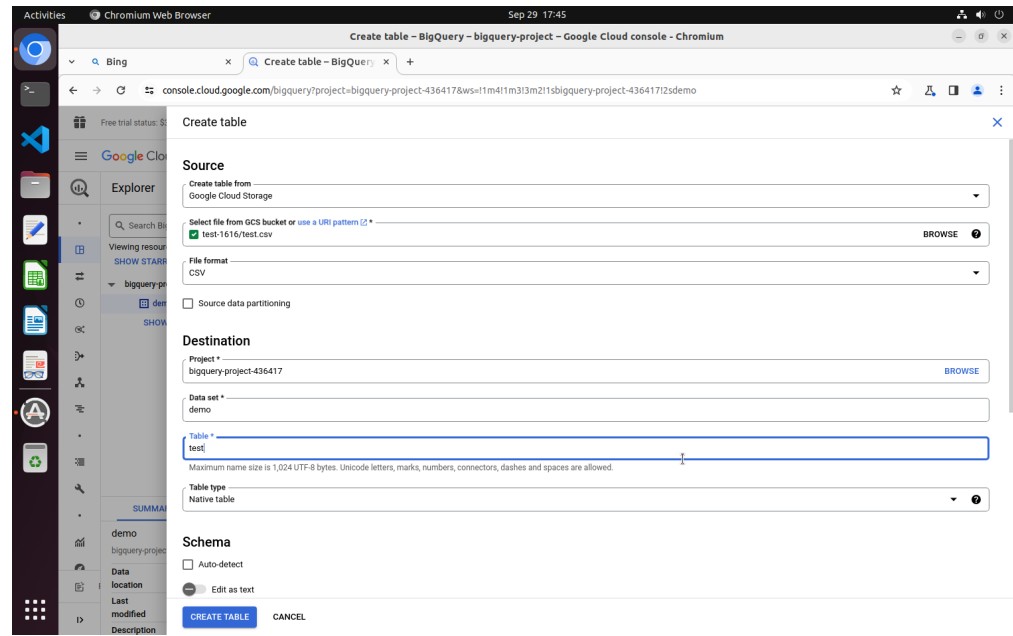

**Action 11**

import pyautogui
pyautogui.click(1425,726) // click to reveal dropdown menu of the table type.

**Observation 11 (Bigquery Interface)**

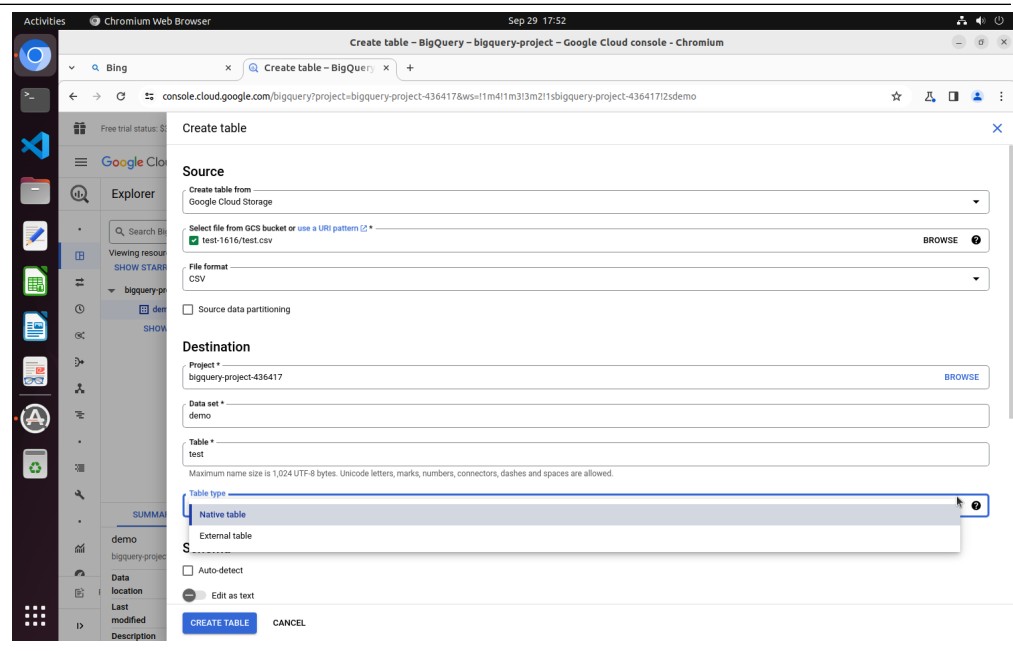

Table 30: **Example of data synthesis - Bigquery**

---

**Action 12**

---

import pyautogui
pyautogui.click(1297,801) **// select the table source external table.**

---

**Observation 12 (Bigquery Interface)**

---

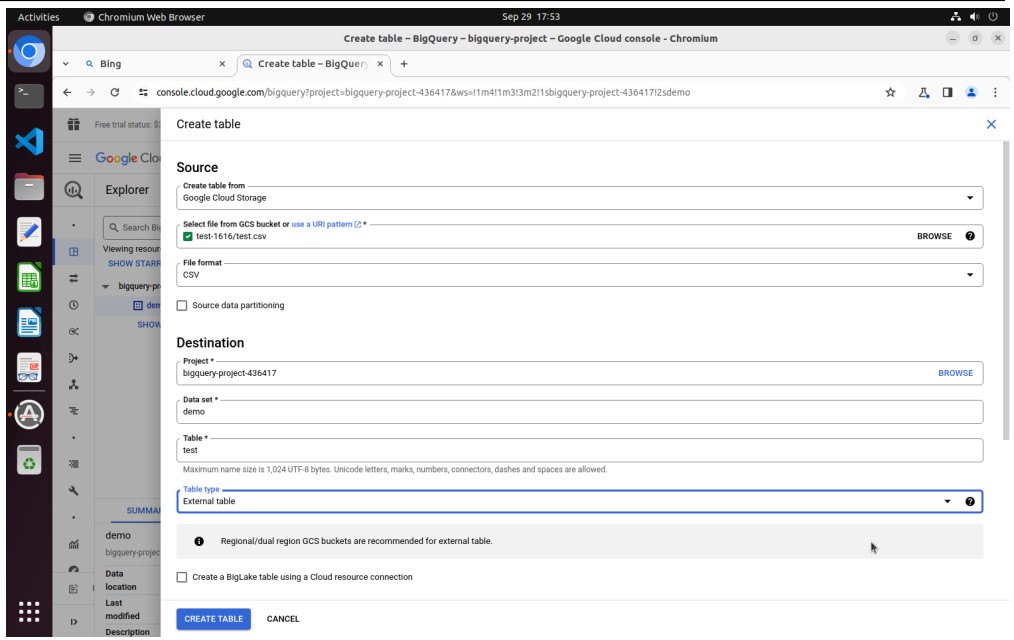

---

**Action 13**

---

import pyautogui
pyautogui.click(303,914) **// click CREATE TABLE.**

---

**Observation 13 (Bigquery Interface)**

---

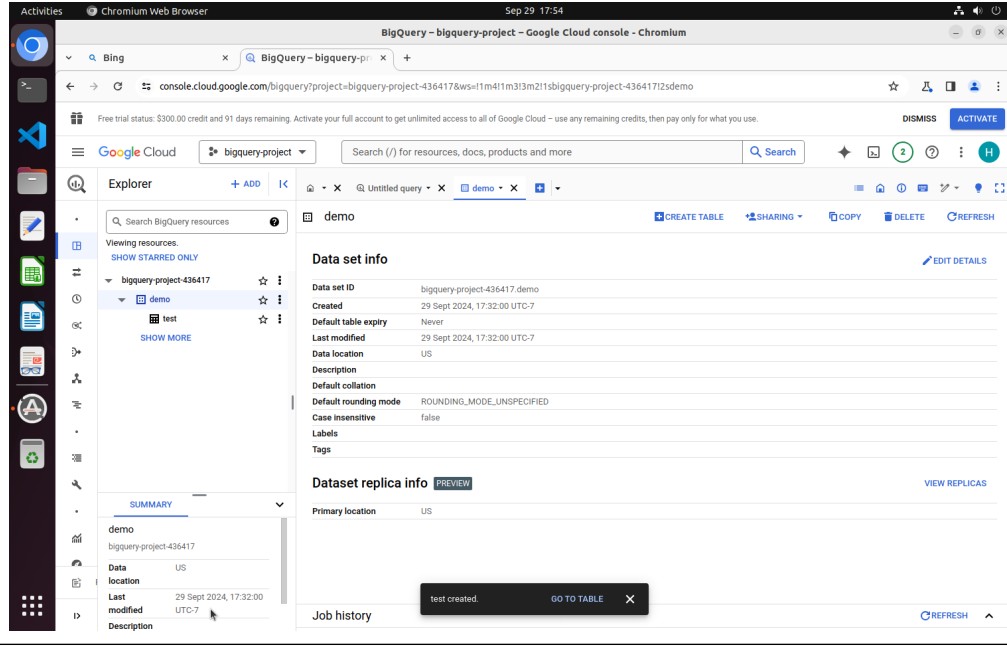

---

Table 31: Instructions generated from trajectory from Table 24 to 30

| sub-trajectory | type | instruction |
|---|---|---|
| Observation 0 ↓ Action 1 ↓ Observation 1) | New task | When is dataset "demo" created? |
| Observation 1 ↓ Action 2 ↓ Observation 2 | Replicate trajectory | Replicate the following: We are currently at the Google Cloud Console interface, specifically focused on a BigQuery project. The browser window displays details of a dataset named "demo" within a BigQuery project. The interface provides information about the dataset, including its creation date, last modified time, data location (US), and other properties like default table expiry and rounding mode. On the left side of the screen, there's a navigation panel showing the Explorer view with the "demo" dataset selected. The top of the screen shows the Google Cloud header with project selection and search functionality. The overall layout is characteristic of a cloud-based data management platform, with options to create tables, share data, and manage dataset properties. After taking the action to click the CREATE TABLE button, we go to the user interface for creating a table. The screen displays a form titled "Create table" with various fields and options. The source section allows selecting a table to create from, while the destination section includes fields for project, dataset, and table name. There's also a schema section and partition and cluster settings. The interface is part of the Google Cloud Console, as evident from the sidebar on the left showing different Cloud services and project navigation. |
| Observation 4 ↓ Action 5 ↓ Observation 5 ↓ Action 6 ↓ Observation 6 ↓ Action 7 ↓ Observation 7 ↓ Action 8 ↓ Observation 8 | New task | Select test.csv in the bucket test-1616 in Google Cloud Storage as the table source. |

Table 32: Instructions generated from trajectory from Table 24 to 30

| sub-trajectory | type | instruction |
|---|---|---|
| Observation 8 ↓ Action 9 ↓ Observation 9 ↓ Action 10 ↓ Observation 10 | Replicate trajectory | Replicate the following: We are in the the interface for creating a table in Google Cloud's BigQuery service. The page is divided into several sections. At the top, it indicates the user is creating a table from a Google Cloud Storage source, with a CSV file selected. The destination section shows the project ID and allows input for the dataset and table name. The destination table is empty. The table type is set to "Native table". At the bottom, there's an option for schema detection, with buttons to create the table or cancel the operation. The left side of the screen displays a navigation menu for the Google Cloud Console, including options like Explorer and various project-related items. The overall layout suggests this is part of a larger cloud data management and analysis platform. After we click on the text box Table, we select and focus on the text box. We then type "test" into the box, which gives the table a name. Except the textbox we are working on, the other parts of the webpage has not changed after clicking and typing. |
| Observation 0 ↓ Action 1 ↓ Observation 1 ↓ Action 2 ↓ ...... ↓ Observation 13 | New task | Link CSV file in Google Cloud Storage to BigQuery |

Table 33: self-instruct prompts to propose instructions based on tutorials, documentations and FAQs.

{Documentation}

Based on the tutorial, examplify 3 tasks that users frequently perform.
User the following format to output:
 ...
 ...

Table 34: Prompts to summarize (sub-)trajectories or propose new tasks based on the (sub-)trajectories.

| **Prompt 1** |
| --- |
| Below is a trajectory to complete a task. |

Observation:
{$Observation_i$}
Action:
{$Action_{i+1}$}
Observation:
{$Observation_{i+1}$}
Action:
{$Action_{i+2}$}
...
Action:
{$Action_{j-1}$}
Observation:
{$Observation_j$}

Please write a reasonable task instruction that is completed by the trajectory.
Wrap the instruction with ``` .

| **Prompt 2** |
| --- |
| Below is a trajectory to complete a task. |

Observation:
{$Observation_i$}
Action:
{$Action_{i+1}$}
Observation:
{$Observation_{i+1}$}
Action:
{$Action_{i+2}$}
...
Action:
{$Action_{j-1}$}
Observation:
{$Observation_j$}

Please summarize the trajectory about each observation and changes after each action.
Wrap the summarization with ``` .

Table 35: LLM prompts to filter low-quality data

Task instruction:
{instruction}
Below is the trajectory to complete the task.
Observation:
{$Observation_i$}
Action:
{$Action_{i+1}$}
Observation:
{$Observation_{i+1}$}
Action:
{$Action_{i+2}$}
...
Action:
{$Action_{j-1}$}
Observation:
{$Observation_j$}

Here are the criteria to indicate a good pair of the instruction and the trajectory:
1. The instruction and the trajectory are aligned, which means the trajectory successfully accomplishes the goal in the instruction.
2. The trajectory is coherent, indicating that each action is logical based on its previous observation and the actions do not contradict with each other based on the task instruction.
3. The trajectory is natural, meaning that the trajectory closely mimics real-world interactions and a human user would possibly perform it when engaging in the environment.
4. The trajectory is reasonable, indicating that the trajectory finishes the task instruction using a reasonable solution, e.g., not using an over-complicated method, not over-simply the problem, not going back and forth in states, etc.

Please answer yes if the task instruction and the trajectory satisfies all the criteria, otherwise, answer with no.

Table 36: Model inference prompts without external knowledge

SYSTEM MESSAGE:
{system message}
OBJECTIVE:
{task instruction}
INTERACTION HISTORY:
{interaction history}
OBSERVATIONS:
{observations}

Your REASONING and ACTION in the format:
REASON:
Your reason to choose a specific action.
ACTION:
Your action

Table 37: Model inference prompts with external knowledge

SYSTEM MESSAGE:
{system message}
ADDITIONAL INFORMATION FOR REFERENCE:
{external knowledge}
OBJECTIVE:
{task instruction}
INTERACTION HISTORY:
{interaction history}
OBSERVATIONS:
{observations}

Your REASONING and ACTION in the format:
REASON:
Your reason to choose a specific action.
ACTION:
Your action

Table 38: Expected model outputs

REASON:

...
ACTION:

...

Table 39: Model prompts to write query for retrieval

SYSTEM MESSAGE:
{system message}
Here is the final goal we want to achieve:
{task instruction}
To achieve the goal, we have done the following:
{interaction history}
Now, we have observed:
{observations}

To better finish the task, write a query to ask for useful information, e.g., what kind of examples or interaction history will be helpful to predict the next action.

Table 40: **OSWorld example (filtered)**

**Instruction**

Sum numbers in the first column.

**Observation 0 (Interface of the software LibreOffice Calc)**

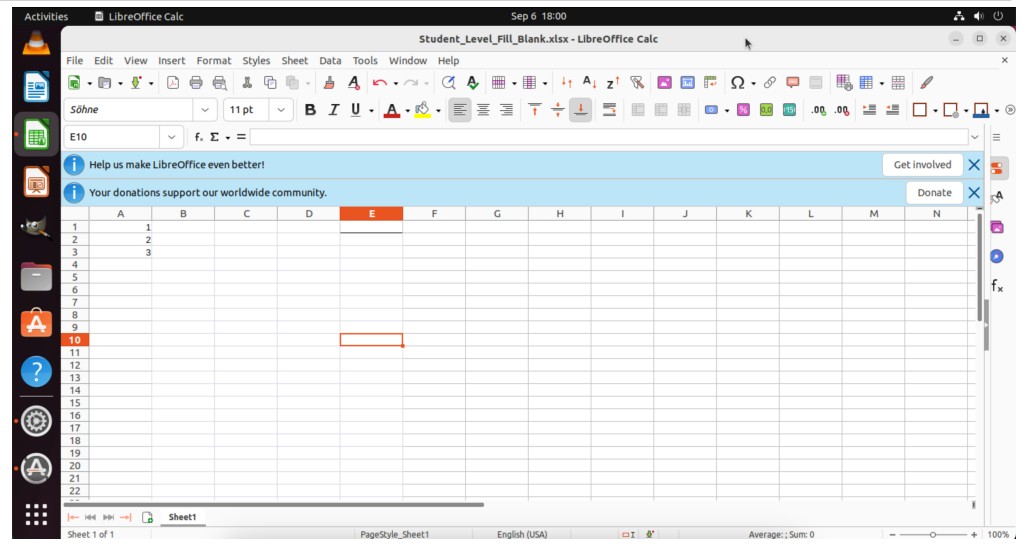

**Action 1**

import pyautogui
pyautogui.click(543,126) // click Tools.

**Observation 1**

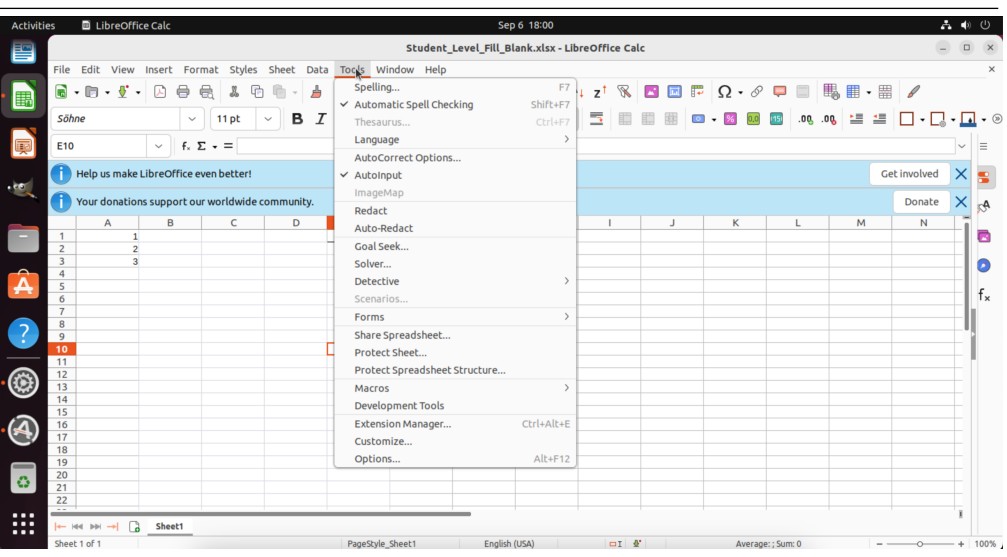

Table 41: **OSWorld example (filtered) cont.**

**Action 2**

import pyautogui
pyautogui.click(543,580) // click Solver.

**Observation 2**

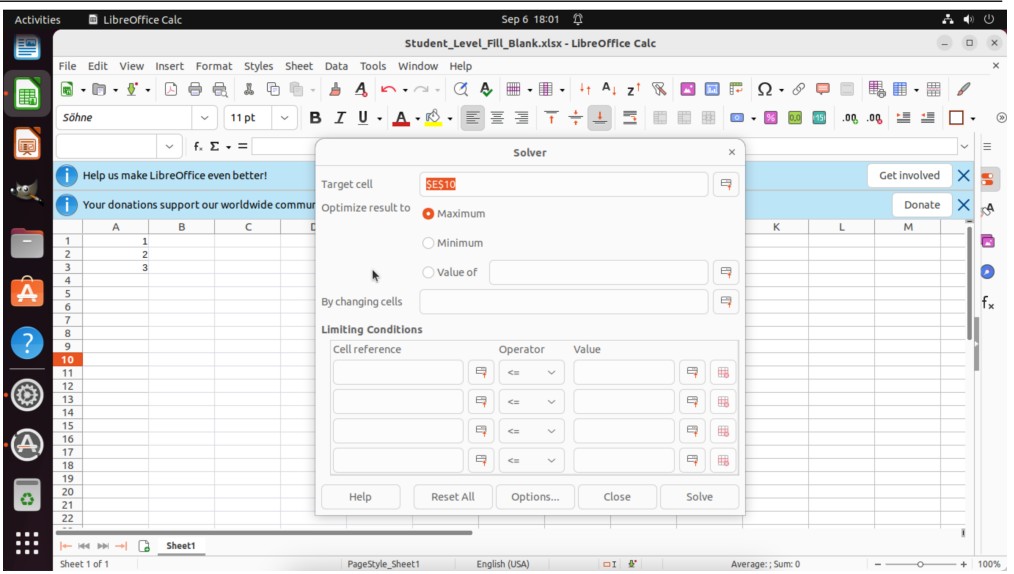

**Action 3**

import pyautogui
pyautogui.click(772,892) // click Close.

**Observation 3**

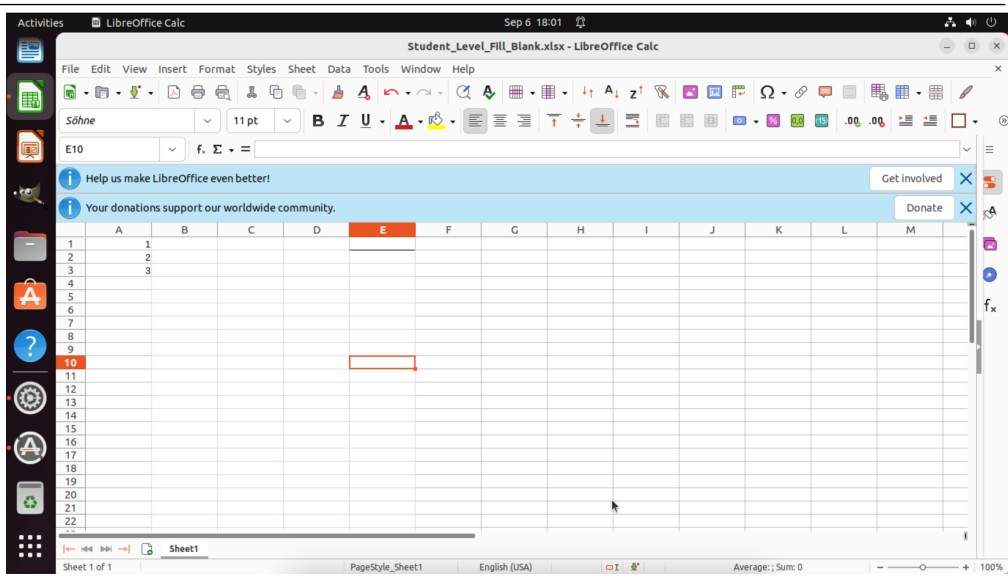

Table 42: **OSWorld example (filtered) cont.**

---

**Action 4**

---

import pyautogui
pyautogui.click(520,126) // click Data.

---

**Observation 4**

---

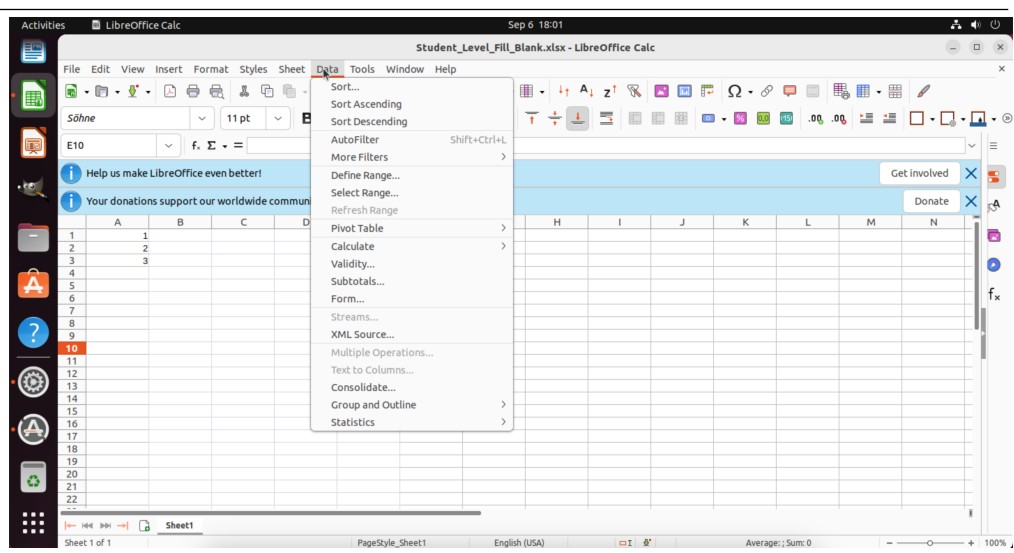

---

**Action 5**

---

import pyautogui
pyautogui.moveTo(520,562) // move to Calculate.

---

**Observation 5**

---

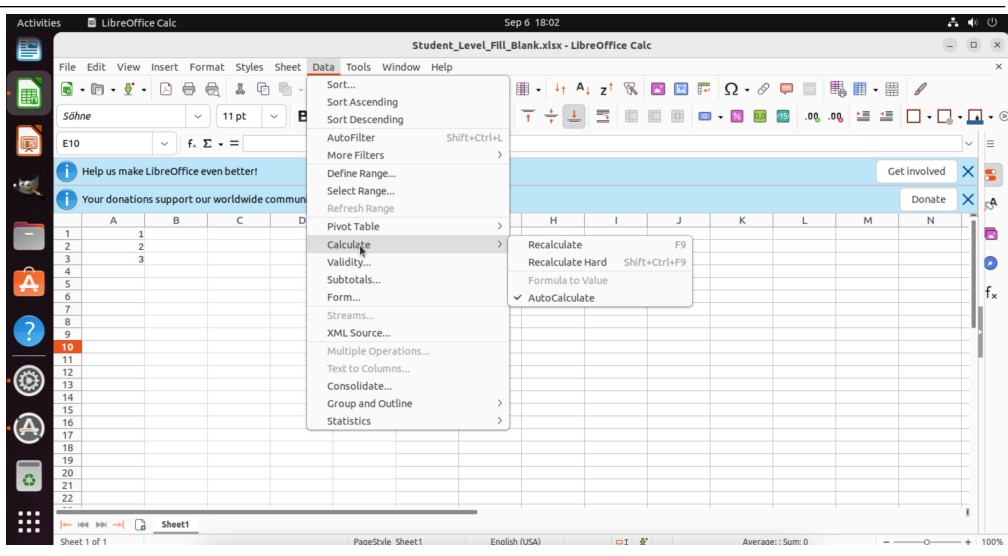

---

Table 43: **OSWorld example (filtered) cont.**

| Action 6 |
| --- |

import pyautogui
pyautogui.click(498,126) // click Sheet.

**Observation 6**

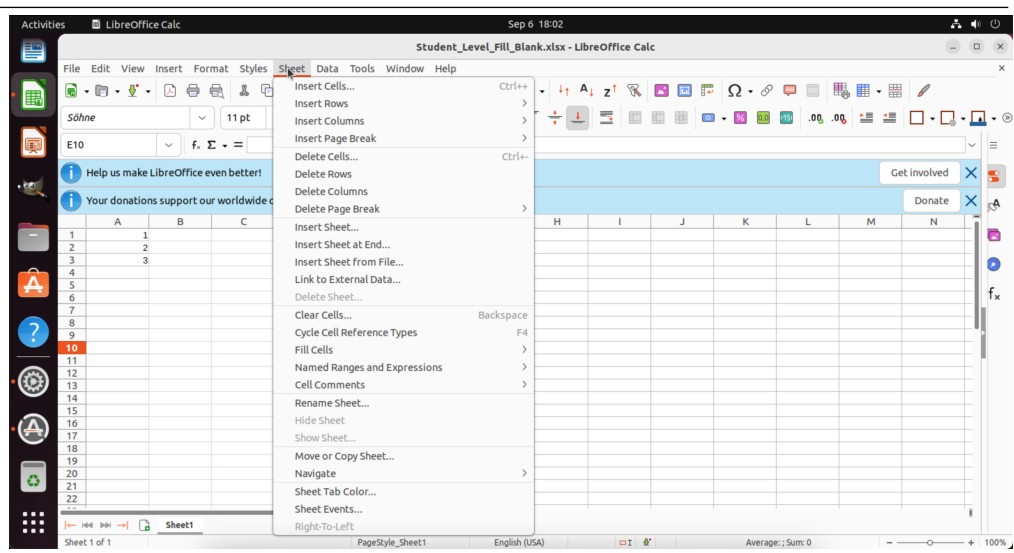

| Action 7 |
| --- |

import pyautogui
pyautogui.click(152,591) // click cell A4.

**Observation 7**

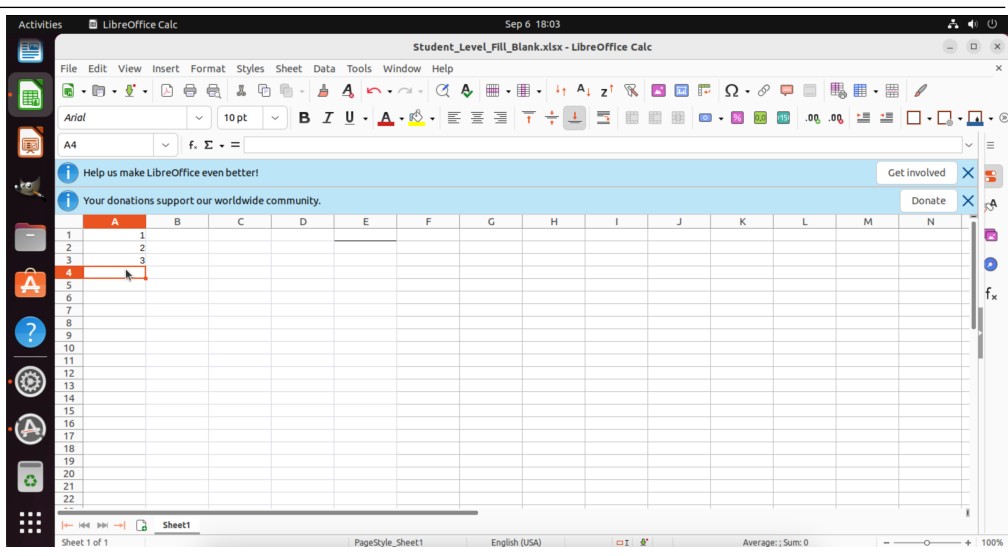

Table 44: **OSWorld example (filtered) cont.**

| **Action 8** |
| --- |

import pyautogui
pyautogui.click(480,302) // click the formula box.

| **Observation 8** |
| --- |

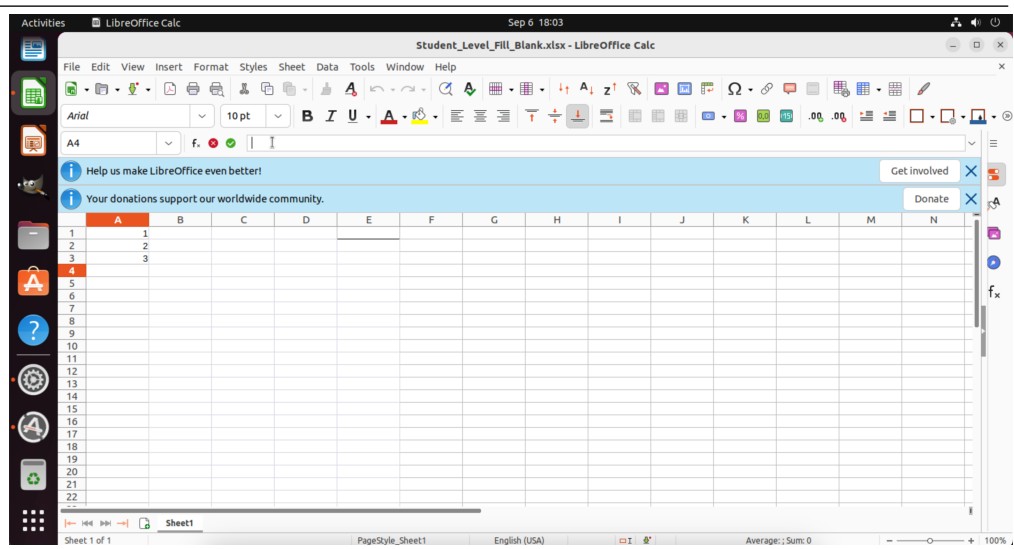

| **Action 9** |
| --- |

import pyautogui
pyautogui.write("=SUM(A1:A3)") // type the formula.

| **Observation 9** |
| --- |

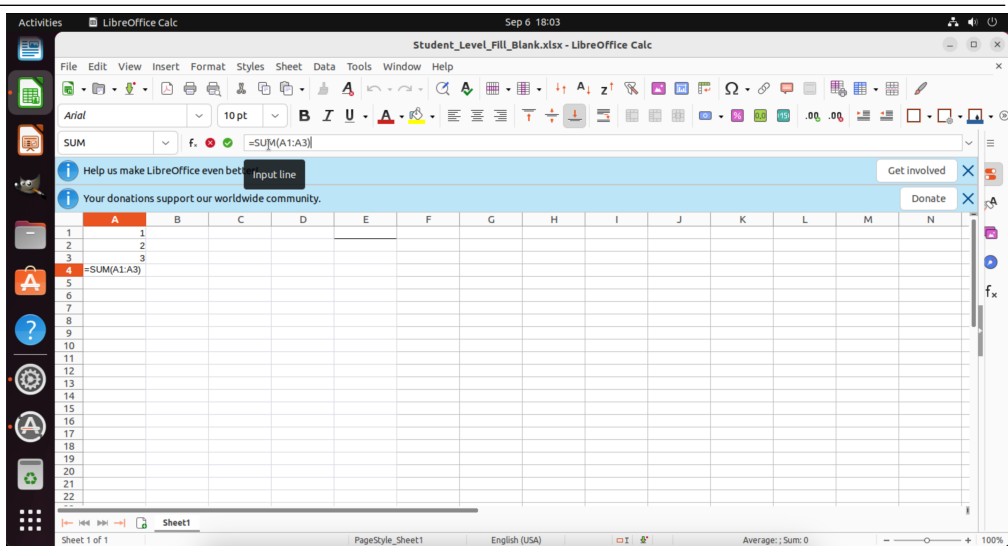

Table 45: **OSWorld example (filtered) cont.**

**Action 10**

import pyautogui
pyautogui.press("enter")

**Observation 10**

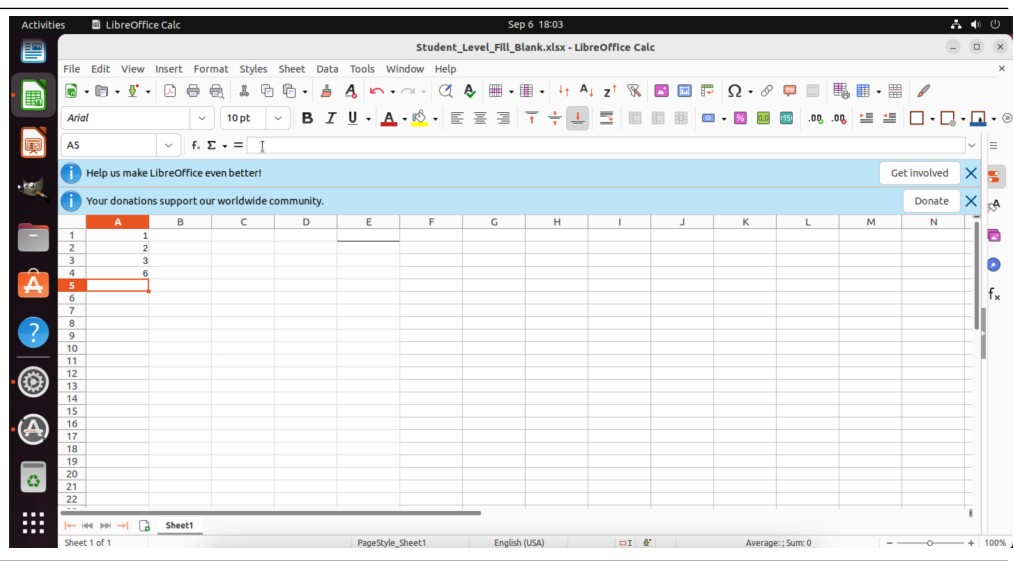

Table 46: **WebArena example (filtered)**

| Instruction |
| --- |
| What are items ordered in the latest cancelled order? |

**Observation 0**

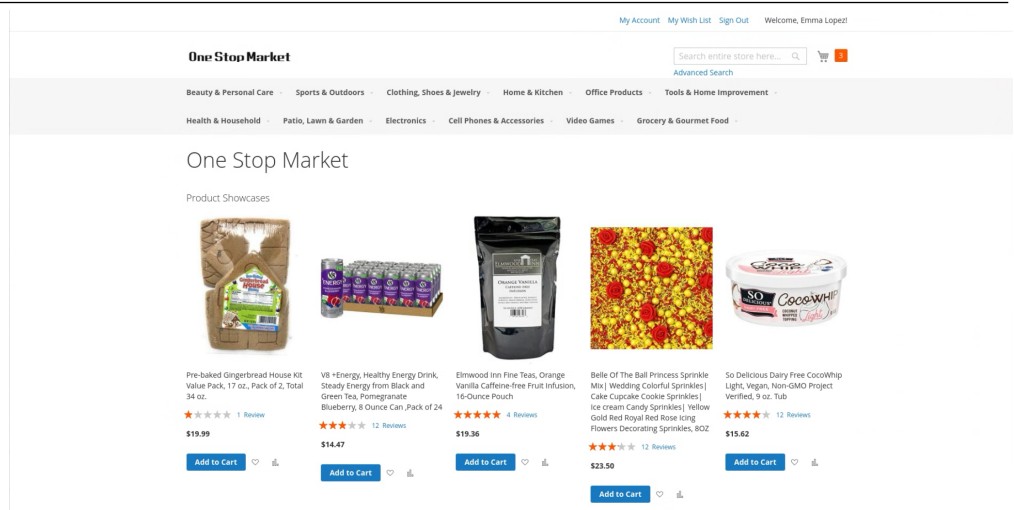

**Action 1**

click [4918] // click the button "My Account"

**Observation 1**

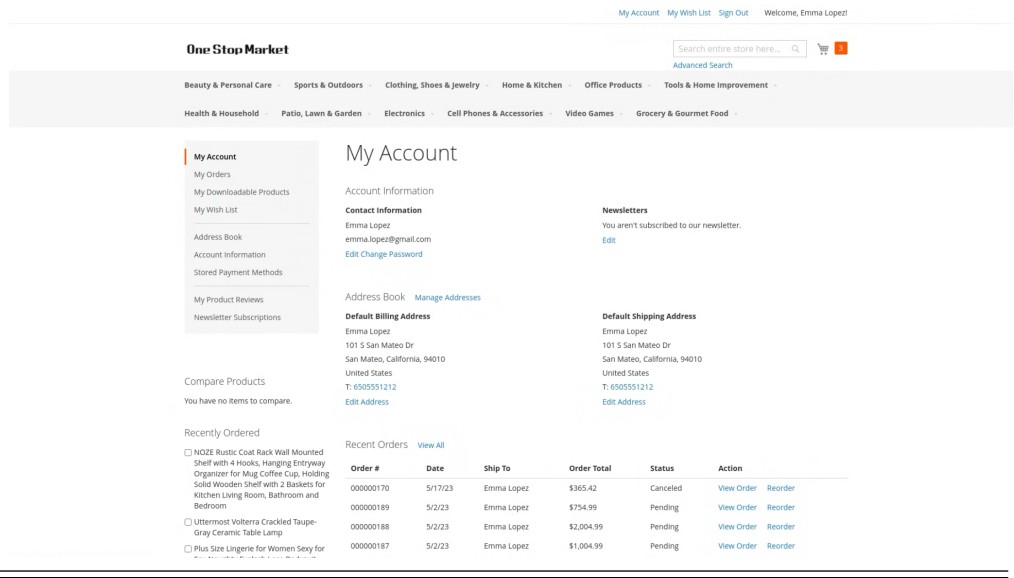

Table 47: **WebArena example (filtered) cont.**

| **Action 2** |
| --- |
| click [4922] // click the button "My Orders". |

| **Observation 2** |
| --- |
| 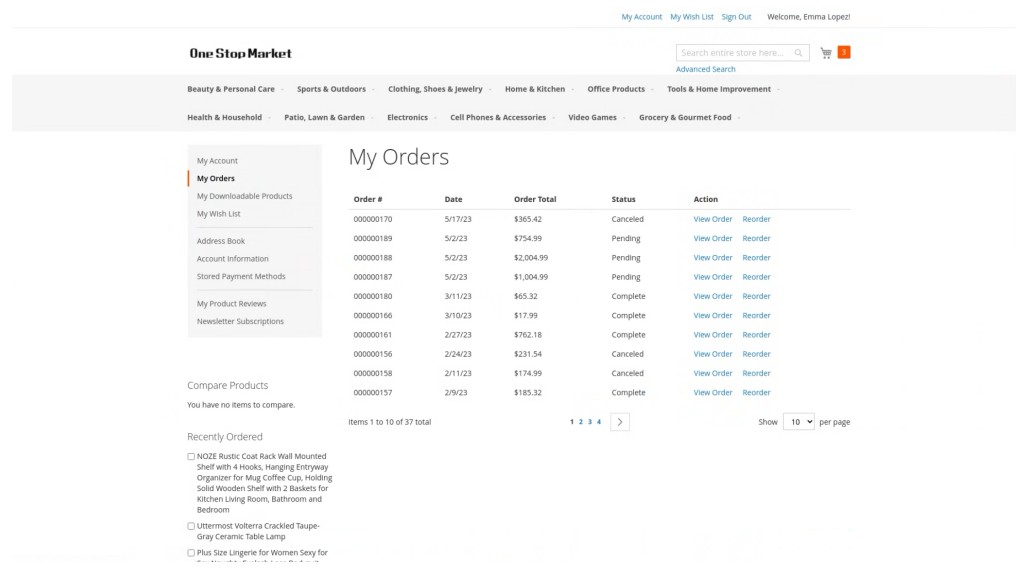 |

| **Action 3** |
| --- |
| click [6357] // click the button "View Order". |

| **Observation 3** |
| --- |
| 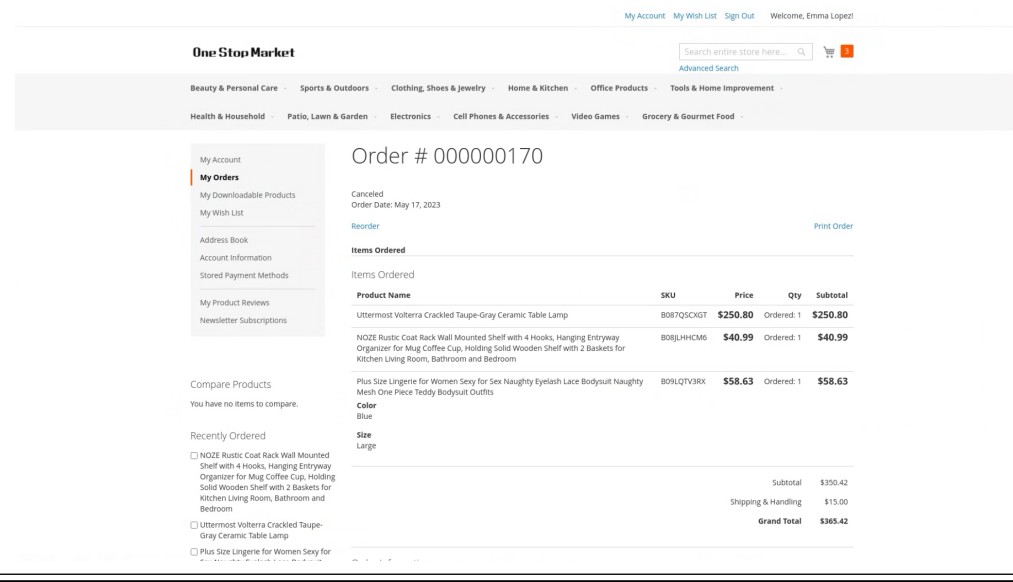 |

Table 48: **WebArena example (filtered) cont.**

| **Action 4** |
| --- |
| click [4922] // click the button "My Orders". |
| **Observation 4** |

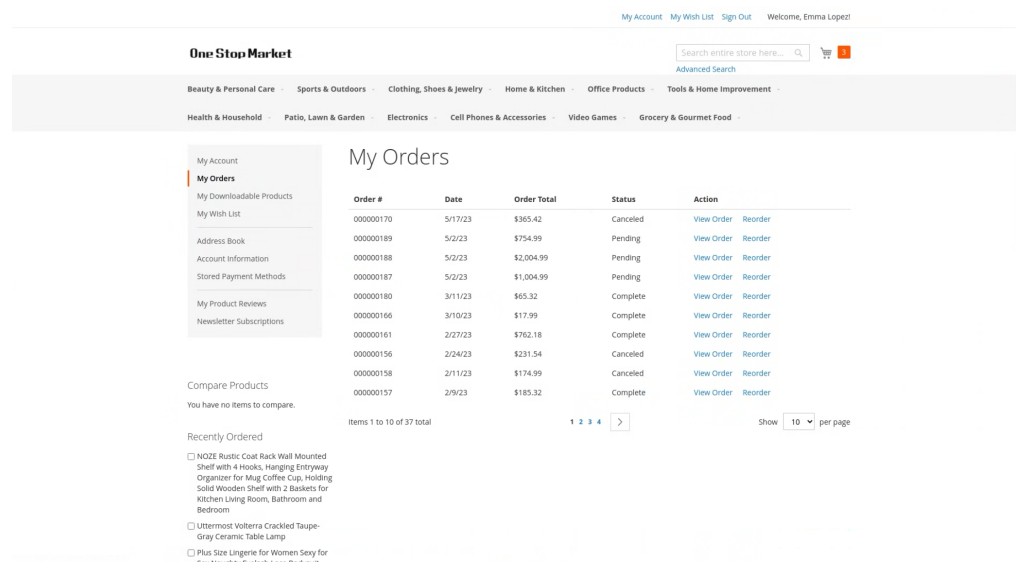

| **Action 5** |
| --- |
| click [6357] // click the button "View Order". |
| **Observation 5** |

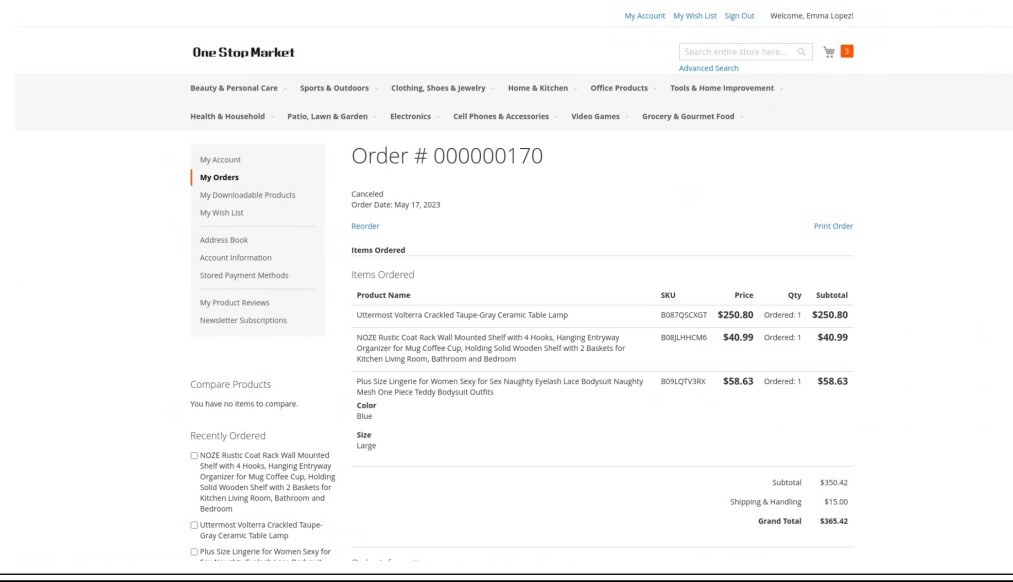

Table 49: **WebArena example (filtered) cont.**

| Action 6 |
| --- |
| click [4922] // click the button "My Orders". |
| **Observation 6** |

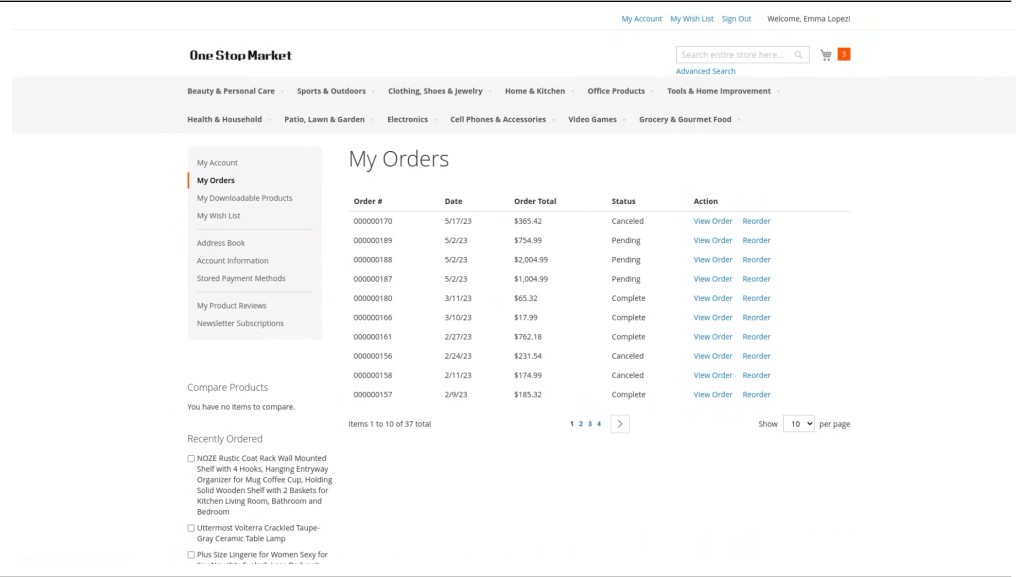

| Action 7 |
| --- |
| click [6357] // click the button "View Order". |
| **Observation 7** |

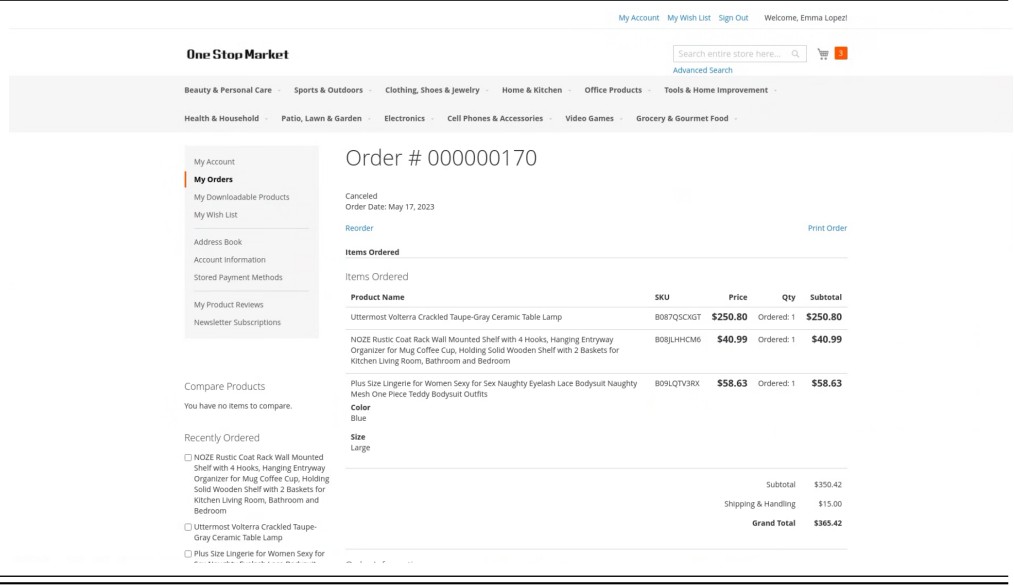

