# OpenReview forum: "Learn-by-interact: A Data-Centric Framework For Self-Adaptive Agents in Realistic Environments"
_ICLR.cc/2025/Conference — ICLR 2025 Poster_

### Official Review · Reviewer_kEqP · 2024-10-28

**Soundness:** 3
**Presentation:** 3
**Contribution:** 2
**Rating:** 6
**Confidence:** 4

**Summary:**

This paper proposes a data synthesis method named “LEARN-BY-INTERACT” that uses LLM to generate instructions and trajectories based on given environmental documents, and the synthesized data can be used in ICL and training scenarios to improve the performance of agents. Experiments conducted on 4 agent datasets validate the effectiveness of LEARN-BY-INTERACT.

**Strengths:**

1. To the best of my knowledge, the backward construction mechanism is novel.
2. The paper is well written.

**Weaknesses:**

1. Although the authors claim that the proposed  LEARN-BY-INTERACT can adapt LLM agents to any given environments without human annotations in both abstract and conclusion sections, but I think its application may not be very wide, it needs the environment to have corresponding documentation, and the LLM used to synthesize the data should be familiar with the environment, otherwise it is difficult for the LLM to synthesize valid instruction and trajectory. More discussion about the limitations of the methodology would make this paper better.

2. There are many works that focus on using more powerful LLMs to synthesize data to improve agent performance, such as AgentGen [1] and AgentTuning [2], but this paper does not discuss or compare them.

3. This method requires a lot of LLM calls to generate and filter the data, especially the backward construction phase, which seems costly.

[1] AGENTGEN: Enhancing Planning Abilities for Large Language Model based Agent via Environment and Task Generation
[2] Agenttuning: Enabling generalized agent abilities for llms.

**Questions:**

Refer to the concerns in “Weaknesses”

---

> ### Author Response · Authors · 2024-11-21
> **Thank you for your review**
>
> We thank the reviewer for recognizing the novelty of the backward construction in synthesizing data and are glad to hear that the reviewer finds the paper well-written. Below we address the concerns raised by the reviewer. We used purple text to clarify the parts that we modified in the revision.
>
> **Weakness 1**:
> First, we would like to clarify that, instead of just in-domain documentation, Learn-by-interact leverages broader resources including software manuals, website FAQs, operation tutorials, and etc. This significantly widens the applications of the proposed approach compared to only considering the documentation. In general, we believe that this setup is a common scenario in realistic environments, as human users frequently use various resources to find solutions. For example, students may look up textbooks to understand a theorem, shopping website users may browse FAQs to check the procedure for returning an item, etc. In this paper, we demonstrate that such resources are available in tens of environments, which indicates the generalizability of Learn-by-interact in practical applications.
>
> Additionally, we do not assume that a pre-trained LLM is familiar with the target environment. Instead, we use a general-purpose LLM and augment it with environment-related resources to generate task instructions. This indicates that a wide range of LLMs could be potentially leveraged to synthesize data following the procedures in Learn-by-interact.
>
> **Weakness 2**:
> We thank the reviewer for pointing out two missing related works in the paper. We add the discussions and citations in section 5. AgentGen focuses on synthetic environments generated by LLMs, while AgentTuning leverages existing datasets and self-instruct to create task instructions. In contrast, Learn-by-interact targets at realistic and complex settings and synthesizes data from scratch based on diverse resources including tutorials, documentation and more.
>
> **Weakness 3**:
> We agree with the reviewer that Learn-by-interact consumes a lot of LLM calls to generate data. However, we would like to clarify: (1). Although the number of LLM calls in backward construction has quadratic complexity in terms of the original trajectory length, it is only linear in terms of the number of generated examples, because each pass of backward construction will correspond to a new instance. (2). In Figure 2, we demonstrate that Learn-by-interact is much more efficient compared to other agentic approaches. It archives significantly higher performance with remarkably fewer LLM calls and consumed tokens. (3). Learn-by-interact is fully autonomous without using any existing datasets or human annotation, which could be even more costly compared to LLM calls in the paper.

---

### Official Review · Reviewer_pZmd · 2024-11-04

**Soundness:** 3
**Presentation:** 3
**Contribution:** 3
**Rating:** 6
**Confidence:** 3

**Summary:**

This paper presents a novel data synthesis framework to enhance agent performance. Contrary to the conventional forward data construction, the proposed backward construction generates instructions from interaction trajectories with the environment. The synthesized data can be used for training the generator or in-context learning. Experiments across four environments demonstrate the potential of this method.

**Strengths:**

In contrast to conventional data synthesis approaches, the proposed backward construction leverages unsuccessful trajectories, thereby improving data collection efficiency. This idea bears a high-level resemblance to the renowned reinforcement learning algorithm Hindsight Experience Replay, which is elegant and proven effective. The paper also provides comprehensive experiments covering performance, efficiency, and scalability.

**Weaknesses:**

However, as shown in Algorithm 1 (lines 16-21), the proposed backward construction has quadratic complexity concerning trajectory length, $O(\text{len}(T)^2)$. This raises concerns regarding data collection efficiency and potentially higher computational costs than conventional forward construction. I am open to raising my score if the authors address the following concerns listed in the Questions section.

**Questions:**

- As mentioned above, the proposed backward construction may have quadratic complexity. I note the relevant discussion in Figure 2, but it is unclear whether this figure applies to inference only or the entire training-inference pipeline.

- On page 3, Algorithm 1 appears to lack a definition of the L() function presented in line 11. Does this function rely on the same LLM backbone as the other function, LLM(), mentioned above?

- On page 5, Table 1, the drop rate is relatively high for OSWorld and Spider2-V, particularly for the latter, where fewer than 20% of synthesized samples are retained. This appears inefficient. Could the authors provide more discussion on this matter?

- Following Question 3, could the authors assess the potential impact of this filtering rate on final performance? For example, if a less strict filtering rule is applied, retaining more samples, how would this affect overall performance?

- In Algorithm 2 on page 4, what is the difference between the append() operation (line 10) and the += operator (line 19)?

- (Minor) It seems that all evaluations lack the percentage unit (%) for accuracy.

---

> ### Author Response · Authors · 2024-11-21
> **Thank you for your review**
>
> Thank you very much for the insightful review! We are glad to hear that the reviewer recognizes the potential of the proposed method and finds that backward construction improves data collection efficiency. Below we address the concerns and questions raised in the review. We used purple text to clarify the parts that we modified in the revision.
>
> **Question 1**:
> We agree with the reviewer that, in terms of the original trajectory length, the complexity is quadratic. However, each pass of backward construction will correspond to a newly generated example. In other words, for an original trajectory with length L=len(T), we need O(L^2) LLM calls to generate task instructions, which produce O(L^2) new instances. Therefore, in terms of the number of synthesized examples N, the complexity of backward construction is linear O(N).
>
> Figure  2 applies to inference only, which illustrates the efficiency of various agentic approaches in downstream applications. We modify the caption to clarify this point.
>
> **Question 2**:
> In Algorithm 1, line 11, the function L refers to LLM, the same LLM backbone. We correct this typo in the revision.
>
> **Question 3**:
> We agree with the reviewer that the drop rate is high in OSWorld and Spider2-V. One explanation attributes this to the increased difficulty in these two benchmarks, as evidenced from the lower baseline performance in Table 2. With harder tasks, we would expect a lower percentage of trajectories to satisfy the criteria to be high-quality data, which leads to a higher drop rate.
>
> **Question 4**:
> Following the reviewer’s suggestion, we re-filter the data in OSWorld and Spider2-V and consider a synthesized instance of high quality if any of LLMs in the committee finds the designed criteria satisfied. We denote this policy as relaxing policy and the one in the paper (require all LLMs in the committee to judge the criteria satisfied) strict policy. The number of synthesized examples before and after two filtering policies are shown below:
> |                          |   OSWorld  |   Spider2-V  |
> | -------------------- | --------------  | ---------------  |
> | Before filtering  |    437,635  |     652,786   |
> | Relaxing policy |    182,788  |     206,323   |
> | Strict policy       |    103,526  |     125,683   |
>
> Using algorithm 2, but replace the data with the version filtered by relaxing policy, we obtain the following results (shown in percentage of the resolved rate):
>
> |                  |   OSWorld           |   Spider2-V          | OSWorld           |   Spider2-V          |
> | -------------- | --------------  | ---------------  | --------------  | ---------------  |
> |    Model   |   Gemini-1.5-pro  |   Gemini-1.5-pro  |   Claude-3.5-sonnet  | Claude-3.5-sonnet  |
> | Baseline                                         |      4.9    |    8.3        |     11.4    |       7.5     |
> | Use data without filtering                 |     6.2    |     11.5      |      14.1   |      11.1    |
> | Use data filtered by relaxing policy |       7.9  |    13.6      |     16.8    |      12.6    |
> | Use data filtered by strict policy      |      10.3  |     16.4     |     22.5    |     16.3    |
>
> We observe a notable performance decrease across two models in both OSWorld and Spider2-V. In particular, using the data filtered by relaxing policy, Claude-3.5-sonnet suffers from a 5.7% drop in OSWorld. This indicates that the additional examples retained by the relaxing policy are of low quality, which makes the overall performance significantly worse.
>
> **Question 5**:
> The append means adding an element to the list, while the operator += means concatenating another list to the current list. For example in Algorithm 2:
> * Before line 10, if R=[[i_1,t_1], [i_2, t_2]], and the new element is [i_3,t_3], then R=[[i_1,t_1], [i_2, t_2],[i_3,t_3]] after line 10.
> * Before line 19, if H=[o_1,a_1,o_2,a_2], and the new list is [o_3,a_3], then H=[o_1,a_1,o_2,a_2,o_3,a_3] after line 19.
>
> **Questioin 6**:
> In Section 3.4, we add a sentence to clarify that all percentage units are omitted for brevity.

---

> > ### Comment · Reviewer_pZmd · 2024-11-24
> >
> > Thank you for providing detailed responses. Most of my comments are addressed.

---

> ### Author Response · Authors · 2024-11-26
> **Thank you!**
>
> Dear Reviewer,
>
> We are glad to hear that most of your comments have been addressed! Thank you very much for your detailed review and constructive feedback on the paper!
>
> Regards, \
> Authors of submission 11615

---

### Official Review · Reviewer_6Xyn · 2024-11-04

**Soundness:** 4
**Presentation:** 4
**Contribution:** 4
**Rating:** 8
**Confidence:** 4

**Summary:**

This paper proposes learn-by-interact, which generates task-specific exemplars which are curated using backward construction which annotates the trajectory with an aligned objective instruction, and filtering using a committee of LLMs.
The results are evaluated on a wide array of benchmarks, SWE-bench, WebArena, OSWorld and Spider2-V, showing the effectiveness over strong baselines.

**Strengths:**

- The proposed approach for generating exemplars for ICL is novel and effective across several agentic scenarios.
- The paper is well written and easy to follow
- The experiments and ablations are very thorough, and validate the components of the proposed method well

**Weaknesses:**

- The discussion/details on filitering of synthesized trajectories could be improved.

**Questions:**

- Can you show examples of what trajectories get filtered out?
- Was LATS implemented by the authors for the benchmarks tested? As far as I'm aware the original LATS didn't evaluate on the benchmarks tested

---

> ### Author Response · Authors · 2024-11-21
> **Thank you for your review**
>
> Thanks a lot for the review! We are glad to hear that the reviewer finds the proposed approach novel and effective across agentic scenarios. We also appreciate the reviewer’s recognition of our paper writing, comprehensive experiments and ablation studies. Below we address the concerns raised by the reviewer. We used purple text to clarify the parts that we modified in the revision.
>
> **Weakness & Question 1**:
> Please refer to our response to the question 3 and 4 of the reviewer pZmd, where we include more discussions on the filtering of the synthesized data and conduct an ablation study to investigate the influence of a lower filtering rate.
>
> In Appendix G, we include representative examples of trajectories that get filtered out.
>
> **Question 2**:
> Yes, we implement LATS on our evaluated benchmarks by ourselves.

---

> > ### Comment · Reviewer_6Xyn · 2024-11-27
> >
> > I appreciate the authors' response, they have addressed my concerns/questions.

---

> ### Author Response · Authors · 2024-11-27
> **Thank you**
>
> Dear Reviewer,
>
> We are delighted to find that we have addressed your concerns and questions! Thanks once again for your insightful comments and suggestions on the paper! They are highly appreciated!
>
> Regards, \
> Authors of submission 11615

---

### Official Review · Reviewer_Hzkc · 2024-11-08

**Soundness:** 3
**Presentation:** 2
**Contribution:** 2
**Rating:** 6
**Confidence:** 4

**Summary:**

This paper aims to address the critical problem of data collection for training language agents. Annotating trajectory-level data in various environments can be quite expensive. To deal with this, this paper instead proposes a data synthesis pipeline that leverages the documentation available on the internet to generate high quality task instructions, and use the LLM to compose the corresponding trajectory for each instruction. Specifically, the error rate of directly generating the trajectory using LLM can be quite high. As a result, this paper proposes a novel scheme called backward construction to summarize the trajectory and refine the original instruction to make it align better with the generated trajectory. In addition, they also use LLMs to filter out low-quality data points. After obtaining the synthetic data, they use them for both fine-tuning and ICL in multiple different domains, including code agent, OS agent, and web agent. Experimental results show the effectiveness of their synthetic data.

**Strengths:**

1. The problem addressed in this paper is highly significant and of great interest to the language agent community. Due to the scarcity of process/trajectory data available online, the community is eager to find efficient and scalable methods to obtain more trajectory-level data for training. The dataset collected in this paper might be a valuable resource to the community.
2. This paper covers multiple domains and demonstrates promising results on all of them, which shows the effectiveness of the data collection process.
3. This paper conducts comprehensive analyses, including scaling law of the training data and the effect of trajectory length.

**Weaknesses:**

1. A key issue with the data synthesis pipeline proposed in this paper is its generalizability. Specifically, the pipeline relies on a set of source documentation to generate meaningful task instructions, serving as the starting point of the entire process. However, the assumption that in-domain documentation will always be available may not hold in all cases.
2. Related to the first point, the reliance on in-domain documentation might also set a ceiling for the size of the dataset. Also, the scaling law in this paper (i.e., Fig 3) suggests that achieving continuous gains becomes challenging once around 100k data samples have been collected.

**Questions:**

Generalizability of this method, for example, for the web domain, it's kinda cheating to use these sources of documentation? WebArena is essentially built on open-source alternatives of these sources. It might be interesting to explore removing the reliance on documentation, as it may not be strictly necessary; maybe you can just ask the LLM to propose task instructions based on the given environment?

In your dataset, is it possible for one data sample to be a sub trajectory of another sample?

On WebArena, why do you choose Step as your baseline method rather than more direct baseline used in the original paper of WebArena?

Typos:
line 44: desktop computing etc. -> desktop computing, etc.
Alg 1: initilize interaction trajectory -> initialize interaction trajectory
Alg 2: Langueg Model -> Language Model

---

> ### Author Response · Authors · 2024-11-21
> **Thank you for your review**
>
> Thanks a lot for the review! We are glad to hear that the reviewer finds the problems addressed in the paper of great significance and that the collected datasets might be a valuable resource to the community. We also thank the reviewer for the appreciation of the promising results and comprehensive analysis of the scaling law and trajectory length. Below we address the concerns raised by the reviewer. We used purple text to clarify the parts that we modified in the revision.
>
> **Weakness 1**:
> First, we would like to clarify that, instead of just in-domain documentation, Learn-by-interact leverages broader resources including software manuals, website FAQs, operation tutorials, and etc. In general, we believe that this setup is a common scenario in realistic environments, as human users frequently use various resources to find solutions. For example, students may look up textbooks to understand a theorem, shopping website users may browse FAQs to check the procedure for returning an item, etc. In this paper, we demonstrate that such resources are available in tens of environments, which indicates the generalizability of the proposed approach in practical applications.
>
> **Weakness 2**:
> We agree with the reviewer that based on the experimental setting in the paper, we observe diminishing returns as the synthesized data size increases. To demonstrate the effectiveness of the synthesized data in a larger scale, we believe that it will depend on joint efforts of model size, architectures, learning algorithms, and more. As shown in Figure 3, one signal indicates that training-based approaches usually achieve more significant improvement compared to the training-free ones, and larger models benefit more by training on synthesized data compared to their small counterparts. It is possible that, in some other settings, e.g., tuning larger models, we can expect more significant gains beyond 100k data as the model has more capacity to incorporate a larger amount of knowledge.
>
> **Question 1**:
> We agree with the reviewer that it is possible to sample task instructions from LLMs based on the given environments. However, we note the following potential concerns regarding this approach: the distribution and the diversity of the generated data are hard to control. Without conditioning on prior documents, one will need intensive prompt engineering to guide LLMs in generating diverse task instructions. On the other hand, the related resources are usually crafted by experts or written by real users, which cover most important scenarios that people who interact with the environment are interested in.
>
> Following the reviewer’s suggestion, we compare Learn-by-interact with the version without relying on existing resources in WebArena. Except for sampling task instructions from LLMs based on given environments, we follow the same procedures in Learn-by-interact to synthesize 10k examples. The results of in-context learning with Claude-3.5-sonnet are shown below:
>
> |  Number of synthesized examples  |   0  |   5k  |      10k  |
> | ------------------------------------------------- | --------------  | ---------------  | ---------------  |
> | Task instructions generated based on environments |  31.5  |    33.0  |     33.6   |
> | Task instructions generated based on related resources |    31.5  |    33.9  |     35.1   |
>
> As shown in the results, using the task instructions only based on given environments results in a performance drop compared to the version that leverages related resources. The gap becomes larger as more data is generated. This indicates the effectiveness of using existing resources to generate high-quality data.
>
> **Question 2**
> Yes, this is possible! The three examples in Table 31 and the first example in Table 32 (in the appendix) can be completed by sub-trajectories of the second example in Table 32. The reviewer may refer to Tables 24-30 for the corresponding visual demonstrations.
>
> **Question 3**:
> In Learn-by-interact, we would like to demonstrate it to be a general approach that can be integrated with many existing methods. Importantly, it is interesting to see that our approach offers additional benefits on top of the state-of-the-art pipeline. This motivates us to choose Step as the baseline implementation at the time we started experiments.
>
> **Question 4**:
> Thanks a lot for pointing out typos in the paper. We have fixed them and highlighted the modifications in purple text in the revised PDF.

---

> > ### Comment · Reviewer_Hzkc · 2024-11-29
> > **thanks for your response**
> >
> > I've decided to raise my score, as most of my concerns are addressed.
> > For web tasks, I think an interesting possibility is to collect more synthetic data by increasing the number of selected websites and see whether we can observe certain cross-website generalization.

---

> ### Author Response · Authors · 2024-11-30
> **Thank you**
>
> Thank you very much! We are glad to learn that most of your concerns have been addressed, and we agree that it would be interesting to investigate the cross-website generalizability of the synthetic data. Motivated by this, we design the following experiments.
>
> Across five websites in WebArena, we consider the content management systems (CMS) as a held-out test set and leverage the synthetic data from the remaining websites as the training data, which includes 81,346 examples. To ensure a fair comparison and avoid the influences of the discrepancies in the training set size, we downsample the original data that covers all the WebArena domains from 109,276 to 81,346 instances. Following the same evaluation pipelines in the paper, we assess the performance of in-context learning with Claude-3.5-sonnet and training with Codestral-22B. The results on the CMS subset are shown below.
>
> |  Model  |  Claude-3.5-sonnet  |   Codestral-22B  |
> | ------------------------------------------------- | --------------  | ---------------  |
> |  Baseline  |    22.0  |     3.3   |
> |  Learn-by-interact with synthetic data that excludes CMS   |   25.2  |    12.6   |
> | Learn-by-interact with all WebArena data that contains CMS  |   28.0  |    17.6   |
>
>
> From the results, we observe that, even without specifically using the data sampled from the CMS environment, Learn-by-interact demonstrates significant improvements over the baseline in both training and in-context learning. This indicates that the proposed approach holds the potential for cross-website generalization and is likely to achieve better performance when utilizing data from more websites.
>
> We hope that this addresses your questions regarding the generalizability of the proposed approach! Thanks once again for your valuable feedback and insightful suggestions on the paper!

---

### Author Response · Authors · 2024-11-21
**Summary response to all reviewers and the new revision**

We thank all the reviewers for their feedback and constructive comments. We are glad to hear that: our approach is effective (Reviewer Hzkc, 6Xyn, pZmd, kEqP); the experiments and the analyses are comprehensive (Reviewer Hzkc, 6Xyn, pZmd); the data synthesis process is novel (Reviewer 6Xyn, kEqP), the studied problem is highly significant and of great interest to the community (Reviewer Hzkc); the paper is well written (Reviewer 6Xyn, kEqP).

In this work, we aim to propose a fully autonomous pipeline to synthesize high-quality agentic data with trajectories. We achieve this by leveraging the backward construction, where we first collect interaction trajectories between LLMs and environments and then synthesize instructions based on them. We demonstrate the effectiveness of the generated data in both in-context learning and finetuning. The evaluation on tens of environments across 4 datasets shows that Learn-by-interact not only significantly improves over the baseline, but outperforms existing agentic approaches by large margins with enhanced efficiency during inference.

In the revision, we updated the draft based on the reviewers’ comments. Updates are denoted in purple text for clarity. Our updates are summarized as follows:
* In the introduction, we add more explanations on our motivation to use various resources including documentation, tutorials, FAQs, etc.
* In section 3.4, we clarify that the percentage unit % is omitted for all performance numbers in the paper.
* We clarify that Figure 2 is only for training-free pipelines during inference in the caption.
* We add more discussions to compare Learn-by-interact with AgentGen and AgentTuning in the related work.
* We add more discussions on the limitations of the proposed approach in section 7.
* We correct 3 typos in the introduction, algorithm 1 and algorithm 2.
* In Appendix G, we add a case study on the examples that get filtered out.

---

### Meta-Review · Area_Chair_3ExS · 2024-12-20

**Metareview:**

This paper proposes LEARN-BY-INTERACT, a data-centric framework that adapts LLM agents to various environments without the need for human annotations. By synthesizing trajectories of agent-environment interactions from documentation and employing backward construction to summarize these histories, the LEARN-BY-INTERACT evaluates the quality of synthetic data in both training-based scenarios and training-free in-context learning (ICL) using innovative retrieval approaches optimized for agents. The paper is well-written, and the authors address almost all of the reviewers' concerns.

**Additional Comments On Reviewer Discussion:**

The paper is well-written, and in the rebuttal phase, the authors addressed almost all of the reviewers' concerns.

---

### Decision · Program_Chairs · 2025-01-22

Accept (Poster)